

# High resolution monthly precipitation isotope estimates across Australia from machine learning

Georgina Falster[1,2], Gab Abramowitz[3], Sanaa Hobeichi[4,3], Cath Hughes[5,6], Pauline Treble[5,6], Nerilie J. Abram[7,8,9], Michael I. Bird[10,11], Alexandre Cauquoin[12], Bronwyn Dixon[13], Russell Drysdale[13], Chenhui Jin[14,15], Niels Munksgaard[10], Bernadette Proemse[16], Jonathan J. Tyler[1], Martin Werner[17], Carol Tadros[5,6]

[1]School of Physics, Chemistry and Earth Sciences, The University of Adelaide, Adelaide 5005, SA, Australia
[2]The ARC Centre of Excellence for Climate Extremes, The Australian National University, Canberra 2601, ACT, Australia
[3]Climate Change Research Centre, UNSW Sydney, Kensington 2052, NSW, Australia
[4]The ARC Centre of Excellence for the Weather of the 21st Century, UNSW Sydney, Kensington 2052, NSW, Australia
[5]ANSTO, Lucas Heights 2234, NSW, Australia
[6]School of Biological, Earth, and Environmental Sciences, UNSW Sydney, Kensington 2052, NSW, Australia
[7]Research School of Earth Sciences, The Australian National University, Canberra 2601, ACT, Australia
[8]The ARC Centre of Excellence for the Weather of the 21st Century, The Australian National University, Canberra 2601, ACT, Australia
[9]The Australian Centre for Excellence in Antarctic Science, The Australian National University, Canberra 2601, ACT, Australia
[10]College of Science and Engineering, James Cook University, Cairns 4878, Queensland, Australia
[11]ARC Centre of Excellence for Indigenous and Environmental Histories and Futures, James Cook University, Cairns 4878, Queensland, Australia
[12]Institute of Industrial Science, The University of Tokyo, Kashiwa 277-8574, Chiba, Japan
[13]School of Geography, Earth and Atmospheric Sciences, The University of Melbourne, Melbourne 3053, Victoria, Australia
[14]ARC Centre of Excellence for the Weather of the 21st Century, Monash University, Melbourne 3800, Victoria, Australia
[15]School of Earth Atmosphere and Environment, Monash University, Melbourne 3800, Victoria, Australia
[16]School of Natural Sciences, University of Tasmania, Hobart 7001, Tasmania, Australia
[17]Helmholtz Centre for Polar and Marine Research, Alfred Wegener Institute, Bremerhaven 27515, Germany

*Correspondence to*: Georgina Falster (georgina.falster@adelaide.edu.au)

**Abstract.** The stable isotopic composition of precipitation ($\delta^2H_P$, $\delta^{18}O_P$; 'water isotopes') is a powerful tool for tracking water through the atmosphere, as well as fingerprinting land-surface water masses and identifying water cycle biases in isotope-enabled climate models. Water isotopes also underpin our understanding of multi-decadal to multi-centennial water cycle variability via their retrieval from palaeoclimate archives. Water isotopes thereby increase our understanding of past and present—and hence future—water cycle variability. Understanding the drivers of spatial and temporal water isotope variability is a critical first step in applying these tracers for a better understanding of the water cycle. However, water isotope observations are sparse in both space and time. Here we develop and apply a machine learning (random forest) approach to predict spatially continuous monthly $\delta^2H_P$ and $\delta^{18}O_P$ across the Australian continent at 0.25° resolution from 1962–2023. We train the random forest models on monthly $\delta^2H_P$ (n = 5199) and $\delta^{18}O_P$ (n = 5217) observations from 60 sites across Australia. We also predict the deuterium excess of precipitation (*dxs*$_P$, defined as $\delta^2H_P - 8*\delta^{18}O_P$). Out-of-sample $\delta^2H_P$ and $\delta^{18}O_P$ prediction skill is high both geographically and temporally. Skill is slightly lower for the secondary





parameter $dxs_P$, likely reflecting the larger reliance of spatio-temporal $dxs_P$ variability on moisture source conditions. The
random forest models accurately capture both the seasonal cycle of precipitation isotopic variability and long-term annual-mean precipitation isotopic variability across the continent, and outperform estimates from an isotope-enabled atmosphere general circulation model over an equivalent time period. We show that spatio-temporal variability in precipitation amount, precipitation intensity, and surface temperature are particularly important for monthly $\delta^2H_P$ and $\delta^{18}O_P$ variations across the continent, with local surface pressure also important for $dxs_P$. Drivers of site-level $\delta^2H_P$, $\delta^{18}O_P$, and $dxs_P$ are more varied.
Overall, the new random forest modelled dataset reveals clear spatial and temporal variability in $\delta^2H_P$, $\delta^{18}O_P$, and $dxs_P$ across the Australian continent over the past decades—providing a robust foundation for hydrology, ecology, and palaeoclimate research, as well as an accessible framework for predicting water isotope values in other locations.

## 1 Introduction

The stable isotopic composition of precipitation ($\delta^2H_P$, $\delta^{18}O_P$[1]) is an integrative tracer of the dynamical processes resulting in a particular amount of precipitation falling in a particular place at a particular time (Bowen et al., 2019; Galewsky et al., 2016). Water molecules comprising only the common light ($^1H$, $^{16}O$) isotopes have higher saturation vapour pressures than water molecules containing a rarer heavy ($^2H$, $^{18}O$) isotope. Consequently, a mass-dependent isotopic fractionation occurs every time water changes phase (evaporation, condensation, sublimation, riming). For example, when water vapour condenses to liquid, water molecules with heavy isotopes will preferentially go to the liquid phase relative to those with light isotopes. The magnitude of equilibrium fractionation depends on the ambient temperature (Majoube, 1970, 1971). Diffusive processes impart additional kinetic fractionation, resulting in important impacts from, for example, ambient relative humidity and the time available for isotopic equilibration between the two phases (Galewsky et al., 2016). Hence, information about any process involving a water phase change is recorded in the stable isotope composition of a moisture parcel as it moves through the atmosphere, and may be preserved in the ultimate isotope composition of surface precipitation. These water cycle processes include (but are not limited to): evaporation from a particular oceanic source (e.g., Araguás-Araguás et al., 1998; Wolf et al., 2020), cloud formation (Aggarwal et al., 2016), sub-cloud evaporation (e.g., Graf et al., 2019; Risi et al., 2010a), and prior rainout along a moisture parcel's trajectory (e.g., Dansgaard, 1964; Gat, 1996; Wang et al., 2020). The derived parameter 'deuterium excess' ($dxs$; $\delta^2H - 8*\delta^{18}O$) describes a water sample's deviation from equilibrium isotope fractionation (Craig, 1961; Dansgaard, 1964). $dxs_P$ provides additional information about temperature and relative humidity at a moisture parcel's source—including any continental moisture recycling (Aemisegger et al., 2014; Pfahl and Sodemann, 2014).

---

[1]There is a third stable oxygen isotope, $^{17}O$. $\delta^2H_P$ and $\delta^{18}O_P$ have been widely measured since the 1960s (e.g., Araguás-Araguás et al., 2000). Observations of 'triple oxygen' ($\Delta^{17}O_P$) are far less common than $\delta^2H_P$ and $\delta^{18}O_P$ due to the difficulty of accurately measuring $^{17}O/^{16}O$ ratios (Aron et al., 2021), and are not dealt with further in this study.





The integrative nature of precipitation isotope tracers means they have applications beyond understanding dynamical variability in the water cycle. For example, water isotopes are useful for fingerprinting terrestrial water masses (e.g., rivers, groundwater) because environmental waters tend to strongly reflect the isotopic composition of the source precipitation (Bowen et al., 2011; Jasechko, 2019; Vystavna et al., 2024). Spatial and temporal water isotope patterns can be further incorporated into flora and fauna that use environmental waters to form tissues or biomolecules containing H and/or O (e.g., cellulose, chitin, lipids, feather, fur, bone, enamel) (Anon, 2008; Hobson, 2023; Meier-Augenstein, 2019). This allows use of spatio-temporal isotope fingerprints for tracking insect and animal migration (e.g., Hobson et al., 2021; Rogers et al., 2022; Wunder, 2012), identifying illegal plant and animal trafficking (e.g., Hopkins et al., 2022; Retief et al., 2014), identifying movement of invasive species (Heinrich and Collins, 2017), food provenancing (Camin et al., 2017; Kelly et al., 2005; Varrà et al., 2023), and the origin and movements of modern and ancient humans—with applications in archaeology, anthropology, and criminal forensics (Bartelink and Chesson, 2019; Depaermentier, 2023; Font et al., 2015; Fraser and Meier-Augenstein, 2007; Mützel Rauch et al., 2009; Obertová et al., 2023).

Water isotope tracers are also used to reconstruct past climates. The stable isotope composition of environmental waters can be incorporated into natural archives with minimal transformation relative to other environmental proxies for climatic variables (Konecky et al., 2020). Palaeoclimate archives that preserve information about their source water isotope composition are globally distributed and include coral and cave carbonate, lake and marine sediments, tree wood, and ice (Konecky et al., 2020)—allowing reconstruction of many aspects of the global water cycle. Water isotope proxy records from these archives have been used to quantify decadal to multi-centennial variability in climate variables ranging from the El Niño-Southern Oscillation (e.g., Falster et al., 2023; Thompson et al., 2013) and the Pacific-North American teleconnection (Liu et al., 2014) to local precipitation amount (e.g., Bird et al., 2020; Hu et al., 2008; Kurita et al., 2016; Sinha et al., 2007; Tierney et al., 2017), groundwater recharge (e.g., Priestley et al., 2023), and dynamical variability in the global water cycle (Konecky et al., 2023).

Water isotope transport and fractionation processes have also been incorporated into the atmospheric water cycle of some general circulation models (GCMs) (e.g., Brady et al., 2019; Hoffmann et al., 1998; Joussaume et al., 1984; Risi et al., 2010b; Steen-Larsen et al., 2017). Understanding water cycle bias in GCMs is important as they are used for both future climate projections and paleoclimate reconstructions. Comparing outputs from isotope-enabled GCMs with precipitation and vapour isotope observations has revealed previously unidentified biases in how these GCMs simulate dynamical variability in the atmospheric water cycle. Identified biases include: model overestimates of deep convection over the mid-latitude oceans (Nusbaumer et al., 2017); too-weak shallow convective mixing in the tropical mid-troposphere (Tanoue et al., 2023); and biases in both mid-troposphere rain evaporation and stratiform rain fraction over India (Nimya et al., 2022).





Understanding the drivers of spatial and temporal $\delta^2H_P$, $\delta^{18}O_P$, and $dxs_P$ variability is a critical first step in applying water isotope tracers for a better understanding of the water cycle. However, $\delta^2H_P$ and $\delta^{18}O_P$ observations are sparse in both space and time (Terzer-Wassmuth et al., 2021). Previous attempts to understand precipitation isotopic variability beyond data from discrete monitoring stations have mostly been via temporally invariant 'isoscapes'. Precipitation isoscapes infer spatially

continuous precipitation isotopic variability between monitoring stations (Bowen, 2010; Bowen et al., 2009). Interpolation is usually performed with a linear geostatistical algorithm that uses information from one or more explanatory variables to predict $\delta^2H_P$, $\delta^{18}O_P$, and/or $dxs_P$ over a particular spatial domain. Explanatory variables are generally geographical, and have commonly included elevation, latitude, and minimum distance to the coast (e.g., Bowen and Wilkinson, 2002)—with the assumption that these variables provide adequate proxies for the dynamical processes affecting a moisture parcel's isotopic

composition. When creating isoscapes, observational data are temporally reduced into long-term mean annual, seasonal, or monthly values. More recently, machine learning approaches have been used to infer spatial variability in global surface seawater $\delta^{18}O$ (Murray et al., 2023) and spatio-temporal variability in precipitation isotopes over New Zealand (Hill et al., 2025) and Europe (Erdélyi et al., 2023; Nelson et al., 2021). In the case of the latter, the machine learning models performed better than both geostatistical models and isotope-enabled GCMs.


The Australian continent has a network of current and former precipitation isotope monitoring stations (Hollins et al., 2018) that has very large gaps in both space and time. This low density of observational precipitation isotope data mirrors the small number of water isotope-based palaeoenvironmental reconstructions for the continent (Konecky et al., 2020), as well as limiting the use of water isotopes in ecological, hydrological, and forensic studies compared with other regions. There are

isoscape estimates of spatially-continuous precipitation variability across the continent (e.g., Hollins et al., 2018; Terzer et al., 2013; Terzer-Wassmuth et al., 2021). However, these isoscapes either 1) do not provide information about precipitation isotope variability through time, or 2) span at most a two-year period, across the subset of the continent with relatively high observational data density (Duff et al., 2025). Here we develop and apply a machine learning (random forest) approach to predict monthly $\delta^2H_P$, $\delta^{18}O_P$, and $dxs_P$ across the entire Australian continent, at 0.25° resolution from 1962–2023. We

quantify the random forest models' out-of-sample predictive skill in both the spatial and temporal domains, and compare the predicted values with outputs from a state-of-the-art isotope-enabled GCM. This allows us, for the first time, to characterise spatio-temporal variability in $\delta^2H_P$, $\delta^{18}O_P$, and $dxs_P$ across the entire Australian continent—and facilitates a wide range of future applications. Outputs are available in netcdf format at monthly and annual resolution from 10.5281/zenodo.15486278 [*will be made public on acceptance of this manuscript*]. We also provide a web app [*link will be made available on*

*publication*] where users can: 1) download $\delta^2H_P$, $\delta^{18}O_P$, and/or $dxs_P$ time series at locations and temporal resolutions of their choice; and 2) obtain maps of locations across the continent matching specific input values (e.g., for sample provenance searches).



## 2 Methods

We used a machine learning approach to model the relationships between each of $\delta^2H_P$, $\delta^{18}O_P$ and $dxs_P$ (the 'target variables') and a range of geographical, meteorological, and dynamical variables (the 'predictors'). The models were then used to predict time series precipitation isotope maps spanning the Australian continent, and to explore the drivers of their predictability across different locations.

### 2.1 Precipitation isotope training data

We trained the models on $\delta^2H_P$ and $\delta^{18}O_P$ observations from 60 unique sites across Australia (Table S1). We also calculated $dxs_P$ for the 59 sites with both $\delta^2H_P$ and $\delta^{18}O_P$ values. The $\delta^2H_P$ and $\delta^{18}O_P$ data are from a mix of published, 'grey', and unpublished sources, as well as the Australian contributions to the Global Network of Isotopes in Precipitation (GNIP) database, which in turn is facilitated by the International Atomic Energy Agency. 'Grey' refers to data published in government reports or student theses. Where water isotope data were sub-monthly, the values were converted to amount-

weighted monthly means using the precipitation amount measurements associated with the water isotope data. If precipitation amount measurements were not available, we used the daily precipitation amount data from the Australian Gridded Climate Dataset v1 (AGCDv1), which is available at 0.05° latitude by 0.05° longitude resolution from 1900 to 2022 (Jones et al., 2009). At some sites there were months with observations from multiple sources (Table S1); in those cases, monthly values were averaged. Temporal coverage at the sites ranged from 2 to 573 months of data, with data in between 1

and 56 calendar years. The dataset spans 1962–2023; this interval was used to set the temporal bounds on our predictions. All site information is summarised in Table S1.

### 2.2 Random forest models to predict precipitation isotopes

  We used random forest regression models to predict monthly $\delta^2H_P$, $\delta^{18}O_P$, and $dxs_P$ values from a suite of 26 geographical, meteorological, and dynamical predictor variables (described in Section 2.3). Random forest models are ideal for capturing

highly non-linear relationships, and handling correlated predictors, and perform well with a small number of target variable samples. These features make random forest models suitable for the target variables and predictors used in this study.

  In brief, our random forest models are ensembles of regression trees (Breiman, 2001). Each tree in the forest is created with a unique bootstrapped (with replacement) subset of the full training dataset. That is, the forest's trees are trained on different but overlapping datasets, introducing randomisation and variability across the forest. Data omitted from the bootstrapped

subset are used by the algorithm to compute the training error and optimise tree construction. In each tree, the training data undergo recursive binary partitioning ('splitting'), with each split at a node defined by a particular threshold in a particular predictor variable. For example, at one node, the precipitation isotope data may be grouped according to whether the amount of precipitation in that month was more or less than 70 mm. At another node, the data may be split according to whether the





site elevation is above or below 20 m. At each node, the predictor variable by which the data are split is chosen from a reduced set of predictor variables randomly selected from the full available set of predictors. From that reduced set, the model chooses 1) the variable (e.g., precipitation amount) and 2) the threshold (e.g., 70 mm) that best homogenise the precipitation isotope samples in subsequent 'child' nodes, reducing the variability of the child nodes compared to the parent node. A different randomly determined subset of predictor variables is used to choose the variable for each split. This binary partitioning continues until the number of samples remaining at a node falls below a threshold. These nodes, which no longer undergo splitting, constitute the leaves of the tree, with their values representing the average precipitation isotope value of the remaining samples. Predictions from all trees in the forest are averaged to provide the final predictions for the given set of predictors.

The random forest models were built using the *ranger* package (Wright and Ziegler, 2017) in R (v4.4.0; R Core Team, 2024). Random forest model hyperparameters were determined objectively using the *tuneRanger* package (Probst et al., 2018). The hyperparameters tuned for each model (with the parameter name in *ranger* shown in italics) were: 1) the size of the reduced set of predictor variables used at each node for splitting (*mtry*); 2) the minimum number of samples in a node to continue splitting (*min.node.size*); and 3) the fraction of the training dataset used in each tree's training dataset (*sample.fraction*). All other model parameters were as per the *ranger* defaults. Random forest models incorporate randomness in both the data subset used to grow the trees, and the subset of variables available for splitting at each node. To account for this inherent randomness, we repeated each of the stages described below 50 times, each with a different random seed. Each isotope system ($\delta^2H_P$, $\delta^{18}O_P$, $dxs_P$) was modelled separately.

### 2.2.1 Quantifying model predictive skill

We tested out-of-sample model skill separately in the spatial and temporal domains (spatial and temporal transitivity). To test spatial transitivity we excluded all data from one site (the 'test site') from the full training dataset, trained a random forest model using the remaining data, then used that model to predict values for the excluded test site. We repeated that process for all sites. To test temporal transitivity we excluded a random 10% of observations from the full training dataset, trained the model on the remaining 90%, then used the model to predict values for the excluded 10%. We repeated that process so every data point was tested out-of-sample.

To quantify skill, we used a suite of seven independent metrics (Table S4 in Abramowitz et al., 2024). The skill metrics are independent in the sense that a change can be made to the predicted values that affects one metric but not the others. The metrics are:

- The Pearson correlation coefficient (r) between the observed and modelled monthly values;
- the Mean Bias Error (MBE), which describes the average size of the deviation between model and observation;





- the Normalised Mean Error (NME), which is the ratio of MBE to the average deviation of observations from the observational mean;
- the difference between the Standard Deviation (SD) of the observed and modelled monthly values (modified from
Abramowitz et al. (2024), who report the absolute difference);
- the difference in the 5th percentile of modelled versus observed values (modified from Abramowitz et al. (2024), who report the absolute difference);
- the difference in the 95th percentile of modelled versus observed values (modified from Abramowitz et al. (2024), who report the absolute difference); and
the density estimate overlap proportion, which summarises the degree of overlap between density estimates calculated for the modelled and observed values (a value of 1 indicates perfect overlap; 0 indicates no overlap).

We report the mean and variance of each skill score across models created with the 50 unique seeds.

### 2.2.2 Estimating predictor importance

Random forest models assess the relative importance of predictor variables based on their impact on the model's predictive skill. This relative importance can be estimated using methods such as permutation and impurity importance. The permutation method involves selecting a predictor variable, shuffling its values, and observing the resulting degradation in model performance based on the out-of-bag error (see Section 2.2) across the entire forest (see Breiman, 2001 for details). Predictor variables that cause a larger degradation in performance are considered more important for accurately predicting

the target variable. The impurity importance method, on the other hand, quantifies the extent to which a predictor variable contributes to reducing the dataset's variability at each split within the forest (Ishwaran, 2015). Variables that result in greater variance reduction have higher impurity importance. It is important to note that relative importance values compare the importance of predictors with each other, rather than being interpreted as absolute measures of importance.

The two methods were tested and yielded similar results; in the interest of brevity we report only results from the permutation method—which is easily interpretable in that in effect, it 'removes' each variable in turn by randomising its values, thereby removing any predictive power. This is also the importance method used by Hill et al. (2025) in their recent assessment of precipitation isotopic variability over New Zealand. We report predictor importance estimates for the final models (incorporating information from all sites). We also report predictor importance estimates for the seven individual

sites with over 200 monthly observations; results are likely not meaningful with fewer than ~200 observations. In all cases, for each predictor variable we report its average rank across the models created with the 50 unique seeds, with 1 being the least important and a rank equivalent to the number of included variables being the most important. Note that the random





forest predictor importance estimation algorithms assign a negative importance value to variables that decrease model skill (this was not the case with any of our chosen predictor variables).

**2.2.3 Building final models**

When creating the final models, we trained the random forest models using the entire water isotope observation dataset. We then used those models to predict monthly $\delta^2H_P$, $\delta^{18}O_P$, and $dxs_P$ values for each 0.25° grid cell over the Australian continent across 1962–2023.

**2.3 Predictor variables**

To predict monthly precipitation isotope variability on a spatially continuous grid across the Australian continent, we assembled a suite of known meteorological, dynamical, and geographical drivers of precipitation isotope spatio-temporal variability (Table S2). Geographical variables include latitude, longitude, minimum distance to the coast ('continentality'), and elevation (of these, only elevation was included in the final models; see Section 2.3.2). Continentality was calculated using the Australian Statistical Geography Standard GDA2020 digital boundary file. Elevation data are from the GEODATA

9 Second Digital Elevation Model Version 3 (Hutchinson et al., 2008), available at 9 second latitude by 9 second longitude resolution (approximately 250 m) over Australia. All meteorological data were derived from the European Centre for Medium-Range Weather Forecasts Reanalysis v5 (ERA5), available at 6-hourly and monthly resolution on a 0.25° latitude by 0.25° longitude grid (approximately 31 km) from 1940 to present (Hersbach et al., 2020; Soci et al., 2024). Meteorological variables include: air temperature, evaporation, fraction of precipitation delivered as snow, mean sea level

pressure, precipitation amount, precipitation intensity, ratio of convective to total precipitation, relative humidity, wind direction, and wind strength (more details provided in Table S2). We note that there is a gridded precipitation amount product specific to Australia that spans our 1962–2023 analysis interval—the AGCDv2 (Evans et al., 2020). However, AGCDv2 is only available at monthly resolution, and daily data are required to calculate precipitation intensity. Nevertheless, we compared monthly precipitation amount in ERA5 with AGCDv2 in discrete locations across Australia and

the two were very similar (not shown). We therefore used the ERA5 precipitation amount data for consistency across the predictor variables.

The source and delivery mechanism of precipitation are important drivers of local precipitation isotopic variability, but extremely difficult to condense into discrete site-level variables. We therefore included 'weather objects' in our suite of

predictor variables, providing a proxy for many of these processes (Table S2). In brief, the synoptic processes responsible for daily precipitation over the Australian continent can be classified into eight weather object categories: anticyclone, cutoff low (troposphere), cutoff low (stratosphere), cyclone, front, potential vorticity streamer, warm conveyor belt (ascent), and warm conveyor belt (inflow)—see Jin et al. (2024) and Sprenger et al. (2017) for details. The weather object/s responsible





for precipitation are implicitly linked to particular weather systems that characteristically transport moisture from distinct

source regions, and follow particular trajectories. Therefore, including the weather object data as predictors incorporates

information about the processes that brought the moisture to its ultimate precipitation location. The precipitation-bringing

weather objects were originally calculated at daily resolution, on a 0.5° latitude by 0.5° longitude grid from 1980–2019 (Jin

et al., 2024).

To include information about possible broader-scale dynamical drivers of Australian precipitation isotopic variability, we

included indices for known remote drivers of Australian precipitation amount. These are: the Niño 3.4 index for the strength

of the El Niño-Southern Oscillation (ENSO); the trans-Pacific sea level pressure gradient (ΔSLP) index for the strength of

the Pacific Walker Circulation; the Dipole Mode Index (DMI) for the strength of the Indian Ocean Dipole; the difference in

the zonal mean SLP between 40°S and 65°S for the Southern Annular Mode (SAM); and the first principal component of

Indian Ocean SST for the Indian Ocean Basin Mode (IOBM). The Nino 3.4 index was calculated as area-mean sea surface

temperature (SST) anomalies in a box 10°S–10°N, 170°W–120°W. The ΔSLP index was calculated as the difference in SLP

anomalies averaged over the central/east Pacific (5°S–5°N, 160°W–80°W) and the Indian Ocean/west Pacific (5°S–5°N,

80°E–160°E) (following Vecchi et al., 2006). The DMI was calculated as the difference in SST anomalies averaged over the

west (10°S–10°N, 50°E–70°E) and east (10°S–0°, 90°E–110°E) Indian Ocean (following Saji et al., 1999). The SAM index

was calculated as the difference in SLP anomalies averaged over 40°S and 65°S (following Velasquez-Jimenez and Abram,

2024). The IOBM index was calculated as the first principal component of linearly detrended SST anomalies over the Indian

Ocean (following Yu et al., 2022). All indices were calculated using SST or SLP data from ERA5 and all area means were

area weighted.

Finally, we explicitly accounted for the seasonal cycle by encoding the month of the year using sine and cosine

transformations and including these as predictors.

**2.3.1 Predictor data processing**

All predictor variable datasets were regridded to a common 0.25° latitude by 0.25° longitude grid using bilinear interpolation

(matching the spatial resolution of ERA5). The weather object data were converted from daily to monthly resolution by

calculating the proportion of total monthly precipitation delivered by each weather object category. To train the models, we

extracted the predictor data from the 0.25° grid cells matching the location of each monitoring site, then filtered the monthly

time series to only retain months with observations. Together with the $\delta^2H_P$, $\delta^{18}O_P$, and $dxs_P$ data, this formed the set of full

training datasets. We retained the full spatio-temporally continuous set of predictor variables for making predictions with the

final models.




The weather object data are available from 1980-2019; all other datasets are either time-invariant or available for the full calendar years matching the precipitation isotope data availability (1962–2023). We therefore performed all methodological steps described in Section 2.2 twice: once using the longer, reduced set of predictor variables (without the weather objects; spanning 1962–2023), and once using the shorter, complete set of predictor variables (including the weather objects; 295 spanning 1980–2019).

### 2.3.2 Predictor variables included in the final models

Previous spatially-continuous estimates of precipitation isotope variability across the Australian continent were predicted based solely on geographical variables (latitude, elevation, continentality). We initially incorporated these as predictors in our random forest models; however, latitude, longitude, and continentality induced visible artifacts in derived metrics (e.g., 300 the slope of the meteoric water line, not shown). Repeating the skill tests (Section 2.2.1) on results from models using a predictor set without these variables did not impact the results, so they were omitted from the final models.

### 2.4 Comparison with observations and precipitation stable isotopic estimates from other sources

We compared final modelled $\delta^2H_P$, $\delta^{18}O_P$, and $dxs_P$ values (Section 2.2.3) with 1) observations from each site; and 2) outputs from an isotope-enabled atmospheric GCM, ECHAM6-wiso, which simulates $\delta^2H_P$ and $\delta^{18}O_P$ (Cauquoin and Werner, 2021). 305 The ECHAM6-wiso simulation had its 3D fields of temperature, vorticity, and divergence, as well as its surface pressure field, nudged toward ERA5 data—and is therefore directly comparable with monthly observations (Cauquoin and Werner, 2021). Outputs from ECHAM6-wiso are available globally at 0.9° latitude by 0.9° longitude resolution from 1979 to 2021 (1979–2018 publicly available, with an additional three years provided by the authors). We assessed both models' (random forest and ECHAM6-wiso) performance in retrieving 1) the full distribution of observed $\delta^2H_P$, $\delta^{18}O_P$, and $dxs_P$ values at each 310 site; and 2) the seasonal cycle of $\delta^2H_P$, $\delta^{18}O_P$, and $dxs_P$ at each site. When calculating the density functions and seasonal cycles, we used isotope values from the time interval in which all three data sources overlap (1979–2021), only including months with observations. In the case of the random forest models, we used predictions from the longer models trained over 1962–2023 with the reduced set of predictor variables (without weather objects), and show the mean of the 50 models created with unique random seeds.

315

We also used both models (random forests and ECHAM6-wiso) to estimate long-term mean precipitation amount-weighted annual mean $\delta^2H_P$, $\delta^{18}O_P$, and $dxs_P$ ('isoscapes'), and compared this with linear regression-based isoscapes (Hollins et al., 2018). The linear regression-based isoscapes were estimated from latitude, altitude, and distance from the nearest coastline, and are available at 0.17° latitude by 0.17° longitude resolution (no time dimension). In all three cases, we estimated the 320 isoscape bias relative to observations by calculating observed long-term mean amount-weighted annual mean $\delta^2H_P$, $\delta^{18}O_P$, and $dxs_P$ at each site with five or more years of observations, then calculating the difference between this observed value and





the matching grid cell in each of the three modelled isoscapes. We report both total and absolute mean bias across all sites—
noting that this is not representative of the true isoscape skill given the temporal and geographical bias in site distribution.
Values are reported as a percentage of the total observed range in mean annual mean $\delta^2H_P$, $\delta^{18}O_P$, and $dxs_P$ at each site, so the
bias estimates for the three isotope systems are roughly comparable. At each site, we only included years with >3 monthly
observations (sites north of 23°S, where it is common for no precipitation to be delivered in dry-season months) or >8
monthly observations (sites south of 23°S, where precipitation delivery is relatively uniform through the year). Noting that
these bias estimates compare average observed values over different time periods: for the random forest and ECHAM6-wiso
models we additionally calculated the absolute and overall bias, only including model years with matching observations.
This test was not possible for the temporally-invariant linear regression-based isoscape.

We note that Duff et al. (2025) used a linear (kriging) approach to estimate monthly $\delta^2H_P$ and $\delta^{18}O_P$ isoscapes across south-
eastern Australia over a two-year time period (2007–2008). However, formal comparison of our random forest results with
the regional monthly $\delta^2H_P$ and $\delta^{18}O_P$ isoscapes of Duff et al. (2025) was not possible as these data are not publicly available.

## 3 Results

### 3.1 Overall model skill

The random forest models' out-of-sample predictive skill across all metrics is high in both the geographical (Figs. 1–2, S1–
S4) and temporal (Figs. 3, S5–S6) domains, indicating that the random forest approach is suitable for modelling precipitation
isotope variability across the Australian continent. All results are robust to the inherent randomness of the method, with
minor differences between the 50 instances of each model. Temporal transitivity is generally superior to spatial transitivity;
that is, the random forest models are better at filling gaps in time than filling gaps in space. There is little difference in
predictive skill between shorter models trained over 1980–2019 (including the weather objects) and the longer models
trained over 1962–2023 (omitting the weather objects). Skill scores for the shorter models are generally slightly better: for
example, the average model-observation correlation coefficient for $\delta^2H_P$ and $\delta^{18}O_P$ spatial transitivity in the shorter models is
0.7 (not shown) compared with 0.68 for the longer models (Figs. 2a, S3a). The exception is the density estimate overlap
proportion for $dxs_P$ with respect to temporal transitivity, which is much higher in the shorter models (including the weather
objects) than the longer models (omitting the weather objects; Fig. S6g).

In terms of predictive skill in modelling isotope variability at out-of-sample locations: for all three isotope systems, there
was no relationship between skill and the distance to the nearest site included in the training dataset (Figs. 1, S1–S2). This
provides confidence that the model predictions for locations with no training data are likely no worse than indicated by the





skill tests at the training data locations—noting that on average, for over 98% of grid cells, >99% of predictor values are within the range of the training dataset (Fig. S7).

355 When predicting values out-of-sample, the random forest models generally under-estimate extreme values at both ends of the distribution. That is, the slope of the linear relationship between observed and modelled values is generally less than 1 (Figs. 4, S8–S9). This is the case for both spatial (Figs. 1–2e-f, S1–S4e-f) and temporal transitivity (Figs. 3e-f, S5–S6e-f), where the modelled 5th percentile values are mostly positively biased, and the modelled 95th percentile values are mostly negatively biased, and standard deviations are mostly lower than in observations. However, the seasonal cycles of $\delta^2H_P$, 360 $\delta^{18}O_P$, and $dxs_P$ are well represented by the random forest models (blue versus grey curves in Figs. 5, S10–S11). This is the case even at sites with very few observations (e.g., rows 7–10 of Figs. 5, S10–S11).

### 3.1.1 Relative predictability of the three isotope systems

Predictive skill for $\delta^2H_P$ and $\delta^{18}O_P$ is similar whilst predictive skill for $dxs_P$ is slightly lower across all metrics. This is the case for out-of-sample skill in both the temporal and spatial domains (Figs. 1–4, S1–S6, S8–S9), for predictions based on the 365 full training dataset (Figs. 5–6, S10–S13), and for the models with the full versus reduced predictor sets. For example, both the mode and shape of site-wise density functions for observed versus modelled $\delta^2H_P$ and $\delta^{18}O_P$ are similar (Figs. 6, S12, grey versus blue curves). This is particularly the case for sites with over ~50 observations (first four rows in Figs. 6, S12) where there is generally a large overlap between the grey (observations) and blue (random forest models) distributions. Additionally, the random forest models accurately reproduce features such as the left-skewed $\delta^2H_P$ and $\delta^{18}O_P$ distributions at 370 the tropical sites (Cairns, Willis Island). In comparison, although the mode of the random forest-modelled $dxs_P$ values is generally accurate at sites with over ~50 observations, the distributions generally have excessive kurtosis (Fig. S13).

### 3.2 Key predictors of $\delta^2H_P$, $\delta^{18}O_P$, and dxs$_P$ spatio-temporal variability

For the final models (incorporating all training data), precipitation amount and precipitation intensity were the most important predictors of $\delta^2H_P$ and $\delta^{18}O_P$ spatio-temporal variability (Fig. 7). Relative humidity, surface temperature, the ratio 375 of convective to total precipitation, and the seasonal cycle were also important. For $dxs_P$, surface temperature and mean sea level pressure were the most important predictors, followed by precipitation amount and relative humidity.

In the shorter models including the weather object data, the weather objects (shown in lavender on Fig. 7) were generally of middling importance, with precipitation delivered by potential vorticity streamers the most important weather object for 380 spatio-temporal variability in all three isotope systems. Remote drivers (e.g., ENSO, SAM, shown in green) tended to be less important across all isotope systems. Of the remote drivers, the ΔSLP and Niño 3.4 indices for tropical Pacific variability were the most important. The fraction of precipitation delivered as snow was the least important predictor of spatio-temporal



variability in all isotope systems—likely because it is only relevant for a very small geographical region. Nevertheless, all predictors contributed meaningfully to the final models. Predictor importance was robust to the inherent randomness of the
models, with minimal differences between the models produced with the 50 unique seeds.

### 3.2.1 Predictor importance for individual sites

Predictor importance for temporal variability in $\delta^2H_P$, $\delta^{18}O_P$, and $dxs_P$ varied by site (Fig. 8). For $\delta^2H_P$ and $\delta^{18}O_P$: precipitation amount, precipitation intensity, relative humidity, and the seasonal cycle were prevalent in the top five most important predictors across sites, but other variables—including those less important for continental spatio-temporal $\delta^2H_P$
and $\delta^{18}O_P$ variability—also scored highly. These include the ratio of convective to total precipitation and local evaporation; weather objects also regularly featured in the five most highly-ranked predictors.

Relative to $\delta^2H_P$ and $\delta^{18}O_P$, there was more inter-site variability in $dxs_P$ predictor importance (Fig. 8 third column). Both surface temperature and precipitation delivered by potential vorticity streamers appeared the most frequently in the five most
highly-ranked predictors across the five sites. Weather objects (shown in lavender) tended to be ranked more highly than for $\delta^2H_P$ and $\delta^{18}O_P$, with local precipitation amount and intensity generally less important.

### 3.3 Spatio-temporal variability in precipitation stable isotopes over the Australian continent

The spatio-temporally complete modelled dataset shows clear interannual variability in all three isotope systems (Figs. 9–10, S14–S15). For one example, the unprecedentedly extreme 2017–2019 eastern Australian 'Tinderbox' drought (Devanand et
al., 2024; Falster et al., 2024) is associated with distinct isotopic anomalies in the annual mean (see also Fig. S16). This is particularly the case for $dxs_P$, where negative $dxs_P$ values associated with the final year of the drought are the most extreme of any in the past 20 years (Figs. S15–S16).

In terms of the total range of values spanned at any one location (i.e., the maximum value occurring at a location minus the
minimum value occurring at that location, across 1962–2023), monthly precipitation isotopic ranges span 30.1–114.3‰, 4.1–17.4‰, and 5.7–30.9‰ for $\delta^2H$, $\delta^{18}O$, and $dxs$, respectively (Fig. 10a–c). Annual-mean site-wise total ranges span 5–61.7‰, 0.5–9.4‰, and 0.8–12.7‰ for $\delta^2H$, $\delta^{18}O$, and $dxs$, respectively (Fig. 10d–f).

### 3.4 Comparison with precipitation isotopic estimates from other sources

### 3.4.1 Monthly variability

In terms of both the seasonal cycle (Figs. 5, S10–S11, grey versus blue and green curves) and the overall distribution of values (Figs. 6, S12–S13), the random forest models generally outperform the physically-based ECHAM6-wiso estimates of



$\delta^2H_P$, $\delta^{18}O_P$, and $dxs_P$ for the equivalent months. For example, the site-wise seasonal cycles of $\delta^2H_P$ and $\delta^{18}O_P$ in ECHAM6-wiso (green curves in Figs. 5, S10) are generally damped (e.g., Yarrangobilly, Mt. Werong), offset (e.g., Cape Grim, Macquarie Marshes), or both (e.g., Melbourne) relative to both observations and the random forest models. The random

forest models perform particularly well compared with ECHAM6-wiso in retrieving the seasonal cycle of $dxs_P$, where the random forest models generally capture both the magnitude and timing of the seasonality, but estimates from ECHAM6-wiso generally have either a larger overall bias (e.g., Margaret River, Lucas Heights) and/or bias in the seasonal cycle magnitude (e.g., Brisbane, Sydney)  (Fig. S11).

Differences in skill between the two models are less distinct in predicting the overall distribution of $\delta^2H_P$ and $\delta^{18}O_P$ values (Figs. 6, S12)—both the random forest and general circulation models perform well. Whilst the distributions of random forest-predicted values are generally closer to observations than those predicted by ECHAM6-wiso, there are sites where ECHAM6-wiso out-performs the random forest models (e.g., Adelaide, Alice Springs). There are also instances where estimates from the two models are similar but both diverge from observations (e.g., Lucas Heights, Mt. Isa). The $dxs_P$

estimates from ECHAM6-wiso lack the positive kurtosis of the random forest estimates (Fig. S13). However, the ECHAM6-wiso estimates have a negative $dxs_P$ bias at most sites; this negative bias is not present in the random forest estimates.

### 3.4.2 Precipitation isoscapes

In terms of the long-term mean annual-mean $\delta^2H_P$ and $\delta^{18}O_P$ across the Australian continent ('isoscapes'; Fig. 11), there are some similarities between estimates from the random forest models, the physical ECHAM6-wiso model (Cauquoin and

Werner, 2021), and linear regressions using geographic variables (Hollins et al., 2018). Common features include: relatively negative values over the Australian Alps, relatively positive values in south-central continental Australia, and a trend to more negative values northward and westward (Fig. 11a–f). The random forest and linear regression isoscapes also predict relatively negative $\delta^2H_P$ and $\delta^{18}O_P$ values over the Tasmanian Alps. The ECHAM6-wiso and linear regression isoscapes predict large areas of very negative $\delta^2H_P$ and $\delta^{18}O_P$ across northern Australia; these are not so strongly present in the random

forest isoscape.

The three $dxs_P$ isoscapes are quite different (Fig. 11g–i). Overall, there are more similarities between the two statistically-derived isoscapes where $dxs_P$ was modelled directly (random forests and linear regressions) than the physically-based isoscape (ECHAM6-wiso) where $dxs_P$ was calculated from $\delta^2H_P$ and $\delta^{18}O_P$. For example, the random forest and linear

regression isoscapes predict relatively positive values over higher-elevation areas (Australian Alps, Tasmanian Alps, MacDonnell Ranges). These features are not present in the ECHAM6-wiso $dxs_P$ isoscape which relies on accurate modelling of the relative variability in $\delta^2H_P$ and $\delta^{18}O_P$.



The random forest $\delta^2H_P$ and $\delta^{18}O_P$ isoscapes are the most similar to long-term mean annual-mean observations; the linear
regression-based isoscape has slightly lower absolute $dxs_P$ bias at the sites/years with observations (mean absolute bias in
Table 1). In the random forest $\delta^2H_P$ and $\delta^{18}O_P$ isoscapes, values tend to be positively biased in central to north-western
Australia, and negatively biased in eastern Australia (Figs. S17–S18). The reverse is broadly true for the ECHAM6-wiso
isoscape; most sites have a negative bias in the linear regression isoscape (Fig. S17, mean overall bias in Table 1). The
random forest $dxs_P$ isoscape has a negative bias at most sites (Figs. S17g, S18f, mean overall bias in Table 1). Overall, the
random forest models' absolute biases are similar across the three isotope systems (Table 1). The ECHAM6-wiso $dxs_P$
isoscape has much larger mean absolute bias than the ECHAM6-wiso $\delta^2H_P$ and $\delta^{18}O_P$ isoscapes; the linear regression $dxs_P$
isoscape has much smaller mean absolute bias than the linear regression $\delta^2H_P$ and $\delta^{18}O_P$ isoscapes (Table 1). When biases are
calculated using only model values in years with observations (as opposed to the full long-term mean), the random forest
models' absolute biases are uniformly lower; the ECHAM6-wiso biases are higher (Fig. S18, Table 1). Nevertheless, the
values are similar to those calculated using all model years, suggesting that these bias estimates provide a reasonable
approximation of skill across the three isoscapes.

## 4 Discussion

Our new spatially and temporally continuous estimates of monthly $\delta^2H_P$, $\delta^{18}O_P$, and $dxs_P$ reveal strong variability across the
Australian continent, both in space (Fig. 11) and through time (Figs. 9–10, S14–S15). Rainfall-related climate extremes (e.g.,
the 2017–2019 Tinderbox Drought; Fig. S16) are reflected in distinct isotopic anomalies, suggesting that dynamical
atmospheric processes associated with extreme climate events could be further interrogated using precipitation isotopic data.
Importantly, we show that even at individual locations, both monthly and interannual variability in the isotopic composition
of precipitation is high (Fig. 10). This has implications for water isotope-based provenance studies, which have often relied
on temporally invariant isoscapes for isotopic fingerprinting (e.g., Fig. 10 compared with Fig. 11).


Exhaustive skill testing suggests that random forest modelling is a robust method for reliably estimating water isotope
variability across the Australian continent in places/times where direct observations are not available. This may include both
sites lacking observations, and times at current or previous monitoring stations where data were not collected. Random forest
models may be particularly useful for the latter, given their relatively high skill in filling time gaps (Figs. 3, S5–S6).

### 4.1 Reasons for mismatch between modelled and observed values

As is common for any predictive model of physical climate variables (e.g., Hill et al., 2025; Nelson et al., 2021), the random
forest models tend to under-estimate extreme values in all three isotope systems (Figs. 4, S8–S9). This may be due to several
factors. First, the most extreme monthly $\delta^2H_P$, $\delta^{18}O_P$, and $dxs_P$ values may be driven by a small number of extreme





precipitation events (e.g., Griffiths et al., 2022; Munksgaard et al., 2012), which ERA5 underestimates compared with
precipitation events closer to the mean (Lavers et al., 2022). Second, we may have missed predictor variables specifically
relevant for the most extreme negative or positive $\delta^2H_P$, $\delta^{18}O_P$, and *dxs*$_P$ values that have also not been previously identified
in the literature or are not practical for inclusion in random forest models. Third, the mismatch in the extremes may be due to
*observational* error, and therefore not well simulated by the random forest models that are mostly trained on data closer to
the median. For example, during low precipitation months, high $\delta^2H_P$ and $\delta^{18}O_P$ and low *dxs*$_P$ values may result from
evaporation during sample collection. Fourth, random forest modelling is an ensemble method where each prediction is the
average of predictions from all trees. This approach enhances robustness to biases by reducing the influence of any single
tree's error. However, it also dampens extremes due to the averaging process, limiting the models' ability to capture outliers.

Regarding the lower skill in simulating *dxs*$_P$ relative to $\delta^2H_P$ and $\delta^{18}O_P$: compared with $\delta^2H_P$ and $\delta^{18}O_P$, *dxs*$_P$ variability is
strongly driven by ambient conditions at the precipitation *source* (Pfahl and Sodemann, 2014). Moisture source conditions
are not well captured in our models, which by necessity rely on largely site-level predictors. The shorter models—
incorporating the weather object data—predicted *dxs*$_P$ more skillfully than the longer models without the weather objects.
Further, the skill increase resulting from inclusion of the weather objects was larger for *dxs*$_P$ than for $\delta^2H_P$ and $\delta^{18}O_P$,
suggesting that the moisture source and transport information inherent in the weather objects is particularly important for
*dxs*$_P$. Nevertheless, these proxies for the moisture source location and conditions were evidently insufficient for fully
capturing spatio-temporal *dxs*$_P$ variability. Future work could attempt to incorporate information about moisture source and
transport path via moisture parcel back-trajectory analysis (e.g., Munksgaard et al., 2012; Stein et al., 2015), although this
would be computationally expensive as calculations would need to be performed for all precipitation events in all months, at
all grid cells (in this case, 732 months for >8,000,000 grid cells).

**4.2 Comparison with linear regression isoscape and isotope-enabled model**

In terms of long-term mean annual-mean $\delta^2H_P$ and $\delta^{18}O_P$, the random forest models have lower absolute bias than linear
regression-based isoscapes using only geographical predictors (Hollins et al., 2018)—despite the random forest models being
calculated at lower spatial resolution (Fig. S17, Table 1). This demonstrates the importance of meteorological and dynamical
variables for precipitation isotope variability across the Australian continent—and that the impact of that variability on
precipitation isotopes is not solely controlled by geography.

One of the major advances of our new random forest models from the linear regression-based isoscape is the addition of the
time dimension. Information about spatio-temporal precipitation isotope variability was previously available from isotope-
enabled GCMs, including the state-of-the-art nudged isotope-enabled ECHAM6-wiso model. Like the random forest models,
ECHAM6-wiso accurately simulates the seasonal cycle and overall distribution of $\delta^2H_P$, $\delta^{18}O_P$, and *dxs*$_P$ values at most sites





(Figs. 5–6, S10–13). However, ECHAM6-wiso predictions of long-term mean $\delta^2H_P$, $\delta^{18}O_P$, and $dxs_P$ are less accurate in high-elevation regions (Figs. 11, S18). This lower accuracy at high-elevation sites is likely due to the lower spatial resolution of ECHAM6-wiso (0.9°) compared with the random forests (0.25°), affecting the model topography (Cauquoin and Werner, 2021). High-elevation regions such as the Australian and Tasmanian alps form the headwaters for catchments important for

both domestic water supply and generation of hydro-electric power (Donohue et al., 2011). The random forests' relatively high skill in modelling high-elevation precipitation isotopic variability will therefore be particularly useful in understanding surface water and groundwater movements in catchments critical for Australia's water security.

**4.3 Drivers of precipitation isotopic variability across the Australian continent**

Australia's precipitation is highly variable both spatially and temporally, with large gradients in precipitation amount and

seasonality across the continent, and large variations from year to year (Nicholls et al., 1997). Our models reveal that this spatio-temporal heterogeneity in Australian precipitation and its intensity is closely linked to spatio-temporal heterogeneity in the isotopic composition of that precipitation (Fig. 7). However, relative predictor importance for monthly $\delta^2H_P$, $\delta^{18}O_P$, and $dxs_P$ variability varies spatially (Fig. 8)—reflecting the large spatial variability in the drivers, sources, and seasonality of Australian precipitation. For example, tropical northern Australia receives most of its rainfall in the austral summer, with

rain generally delivered by monsoon troughs and tropical cyclones (Sharmila and Hendon, 2020; Suppiah, 1992). Moisture in the monsoon season mostly originates from the proximal seas north of Australia, with up to ~11% local recycled precipitation (Holgate et al., 2020). In comparison, much of southern Australia has winter-dominated precipitation, with moisture generally delivered by extratropical cyclones, fronts, and thunderstorms, and minimal precipitation recycling (Holgate et al., 2020; Pepler et al., 2020, 2021). Inland Australia is mostly arid, with highly variable precipitation (Van

Etten, 2009), delivered by a wide range of weather systems (Acworth et al., 2016), and with moisture sourced from oceans all around Australia as well as terrestrial recycling (Acworth et al., 2024; Holgate et al., 2020). South of the tropics, moisture sources to the Australian continent vary widely, with the Coral and Tasman seas supplying much of the south-eastern to eastern coasts, and the proximal Indian and Southern oceans supplying south-western Australia (Holgate et al., 2020). The timing and source of precipitation over the Australian continent is influenced by both local weather systems and remote

drivers (Risbey et al., 2009)—including ENSO (McBride and Nicholls, 1983), the IOD (Ummenhofer et al., 2009), and the SAM (Hendon et al., 2007).

Accordingly: at Darwin in tropical northern Australia, mean sea level pressure, rainfall amount, and rainfall intensity are in the top five most important predictors of variability in all three isotope systems (Fig. 8 top row). This is likely linked to the

monsoon troughs typically associated with the intense rainfall delivered during 'active' phases of the austral summer monsoon season—and which are associated with moisture that generally comes from the same source region and follows the same trajectory (Berry and Reeder, 2016). Rain delivered during 'inactive' phases of the monsoon is typically not associated




with a monsoon trough, and results from a different range of circulation and moisture flux regimes (Godfred-Spenning and Reason, 2002). In contrast, at Cape Grim in south-eastern Australia (Fig. 8 bottom row): whilst precipitation amount and

intensity are important for precipitation isotopic variability, the particular weather systems delivering that precipitation—including fronts, anticyclones, and potential vorticity streamers—are also important. This is likely due in part to these systems bringing precipitation along different trajectories from different ocean sources.

**4.4 Applications**

Our analyses reveal distinct spatial and temporal variability in $\delta^2H_P$, $\delta^{18}O_P$, and $dxs_P$ across the Australian continent. This

isotopic heterogeneity—combined with the random forest models' unprecedented out-of-sample skill, high spatial resolution, and monthly temporal resolution over a span of 62 years—provides a strong foundation for future Australia-focussed hydrological, ecological, and archaeological research (e.g., Adams et al., 2022, 2023; Bunney et al., 2023; Gibson et al., 2008; Keegan-Treloar et al., 2024; McInerney et al., 2023; Theden-Ringl et al., 2011), as well as food provenancing (e.g., Anh et al., 2022; Simpkins et al., 1999) and forensic investigations (e.g., Jones et al., 2016; Smith et al., 2022). Further, site-

level research previously restricted to using precipitation isotopic data from the nearest GNIP station—often over >100 km away (e.g., Banks et al., 2021; Buzacott et al., 2020; Zhou et al., 2022)—can now incorporate site-specific estimates from our spatio-temporally continuous random forest model outputs.

The new $\delta^2H_P$, $\delta^{18}O_P$, and $dxs_P$ maps provide an unprecedented opportunity for quantifying the particular hydroclimatic

signal/s preserved in water isotope proxy records (e.g., better defining modern isotope-climate relationships), which can then be reconstructed over preceding centuries to millennia. This will be especially valuable for input to proxy system models (Dee et al., 2015; Evans et al., 2013), which explicitly resolve the range of environmental processes by which a precipitation isotopic signal is encoded in a particular palaeoclimate archive (e.g., tree wood (Roden et al., 2000), lake sediment (Dee et al., 2018), cave carbonate (Hu et al., 2021)). Proxy system models can provide detailed insights into the drivers of temporal

variability in a water isotope proxy record—and ultimately more accurate palaeoclimate reconstructions—but generally require estimates of precipitation isotopic variability as inputs.

**Conclusions**

We developed a dataset of monthly $\delta^2H_P$, $\delta^{18}O_P$, and $dxs_P$ values at 0.25° spatial resolution from 1962–2023 over the entire Australian continent, using a random forest modelling approach. The dataset provides unprecedented insights into

precipitation isotope variability in both space and time, as well as the drivers of that variability. The precipitation isotope data are predicted using a freely available suite of meteorological, geographical, and dynamical variables. Additional models spanning 1980–2019 include a set of 'weather object' data, but the predictive skill increase from addition of these additional



predictors is mainly restricted to $dxs_P$. Our preliminary analyses reveal distinct spatial and temporal variability in $\delta^2H_P$, $\delta^{18}O_P$, and $dxs_P$ values across the Australian continent, with applications in many fields. Future work will focus on quantifying the nature and drivers of spatio-temporal variability in Australian precipitation isotope variability.

We show that random forest modelling provides an accurate and inexpensive means of estimating missing $\delta^2H_P$, $\delta^{18}O_P$, and $dxs_P$ values both in space and through time. Prediction errors relative to observations are lower than existing predictive tools. For an area the size of the Australian continent—and using a carefully-selected suite of predictor variables—random forest models can be built and tested on a personal computer using open-source software. Our methods therefore provide an accessible framework for predicting water isotope values in other locations with sufficient observational data density.

**Data Availability**

The continent-wide modelled precipitation $\delta^2H_P$, $\delta^{18}O_P$, and $dxs_P$ values produced in this study are available in netcdf format from the following Zenodo repository: 10.5281/zenodo.15486278 [*will be made public on acceptance of this manuscript*]. The data are available at monthly or annual resolution, from both the shorter models trained over 1980–2019 (including the weather objects) and the longer models trained over 1962–2023 (omitting the weather objects).

Users can view and download (as csv) time series of modelled precipitation $\delta^2H_P$, $\delta^{18}O_P$, and $dxs_P$ values at locations of their choice from the following website: [*link to web app to be provided on publication*]. In this case, the data come from the models trained over 1962–2023 (omitting the weather objects). At the same website, users can also produce maps of locations where particular $\delta^2H_P$, $\delta^{18}O_P$, and $dxs_P$ values occur over a specified time window.

**Author contributions**

Conceptualization, Data curation, Formal analysis, Funding acquisition, Project administration, Validation, Visualisation, Writing (original draft preparation): GMF. Data collation, Early concept discussions: GMF, CH, PT. Methodology: GMF, GA, SH. Resources (providing unpublished data): GMF, CH, NJA, MIB, AC, BCD, RD, CJ, NM, BP, JJT, MW. Resources (providing published data): CT. Writing (review and editing): GA, GMF, SH, CH, PT, NJA, MIB, AC, RD, JJT, CT.

**Competing interests**

The authors declare that they have no conflict of interest.



## Acknowledgements

GMF, CJ, GA, NJA, and SH were supported by the Australian Government through the Australian Research Council (ARC)
Centre of Excellence for Climate Extremes (CE170100023). GMF was also supported by an ARC Discovery Early Career
Researcher Award (DE250100071) and an Australian Institute of Nuclear Science and Engineering (AINSE) Early Career
Researcher Grant (ALNGRA2205). CJ, NJA, and SH acknowledge support from the ARC Centre of Excellence for the
Weather of the 21st Century (CE230100012). MIB acknowledges support from the ARC Centre of Excellence for Australian
Biodiversity and Heritage (CE170100015) and an ARC Laureate Fellowship (FL140100044). JJT acknowledges support
from an ARC Future Fellowship (FT230100648). BCD was supported by an Australian Postgraduate Award and an AINSE
postgraduate award. The Australian GNIP program is funded and coordinated by the Australian Nuclear Science and
Technology Organisation (ANSTO) and supported by the Bureau of Meteorology (BOM), the Commonwealth Scientific and
Industrial Research Organisation (CSIRO) and the International Atomic Energy Agency. We acknowledge contributions
from many past and present ANSTO employees for GNIP coordination and sample analysis, CSIRO Land and Water
Adelaide for sample analysis prior to 2005, and BOM station operators for collecting samples. Jagoda Crawford and Alan
Griffiths contributed to GNIP data compilation and quality control. This research was undertaken with the assistance of
resources from the National Computational Infrastructure, an NCRIS enabled capability supported by the Australian
Government. We thank the European Centre for Medium-Range Weather Forecasts for the ERA5 reanalysis data sets. The
ECHAM6-wiso simulation was performed at the Alfred Wegener Institute supercomputing centre.

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



**Tables and figures**

**Table 1: Modelled isoscape $\delta^2H_P$, $\delta^{18}O_P$, and *dxs*$_P$ bias with respect to observed long-term means**. Only including sites with five or more years of observations. The values state the average bias across all sites (shown in Figs. 10, S17). All bias estimates are shown as a percentage of the total range of variability in that particular isotope system ($\delta^2H_P$, $\delta^{18}O_P$, or *dxs*$_P$). Values in parentheses were calculated using the subset of values present in *both* the random forest and ECHAM6-wiso datasets, to facilitate direct comparison between the two (shown in Fig. S18). Rows in the top half of the table show the absolute bias values; rows in the bottom half of the table show the overall bias (i.e., highlighting the average direction of each product's bias). Note that *dxs*$_P$ was modelled directly in the random forest and linear regression models. For ECHAM6-wiso, *dxs*$_P$ was calculated from the simulated $\delta^2H_P$ and $\delta^{18}O_P$ and the bias values are shown in italics. To facilitate fair comparison, we show the average absolute bias for only $\delta^2H_P$ and $\delta^{18}O_P$ (estimated directly in all three products) as well as the average absolute bias across the three isotope systems. .

| | Random forests | ECHAM6-wiso | Linear regression |
|---|---|---|---|
| | *Mean absolute bias (%): all years* | | |
| $\delta^2H_P$ | 8.8 (7.4) | 15.6 (19) | 16.4 |
| $\delta^{18}O_P$ | 7.1 (5.8) | 13 (15.5) | 13 |
| *dxs*$_P$ | 8.2 (7.1) | *25.6 (27)* | 6.9 |
| **Average ($\delta^2H_P$ & $\delta^{18}O_P$)** | **8.0 (6.6)** | **14.3 (17.2)** | **14.7** |
| **Average (all)** | **8.0 (6.8)** | *18.1 (20.5)* | **12.1** |
| | *Mean overall bias (%): all years* | | |
| $\delta^2H_P$ | -1 (-0.7) | -3.4 (-6.6) | -15.4 |
| $\delta^{18}O_P$ | -0.1 (-0.4) | 0.7 (-1.3) | -11.9 |
| *dxs*$_P$ | -6.6 (-5.3) | *-18.7 (-20.2)* | 1 |



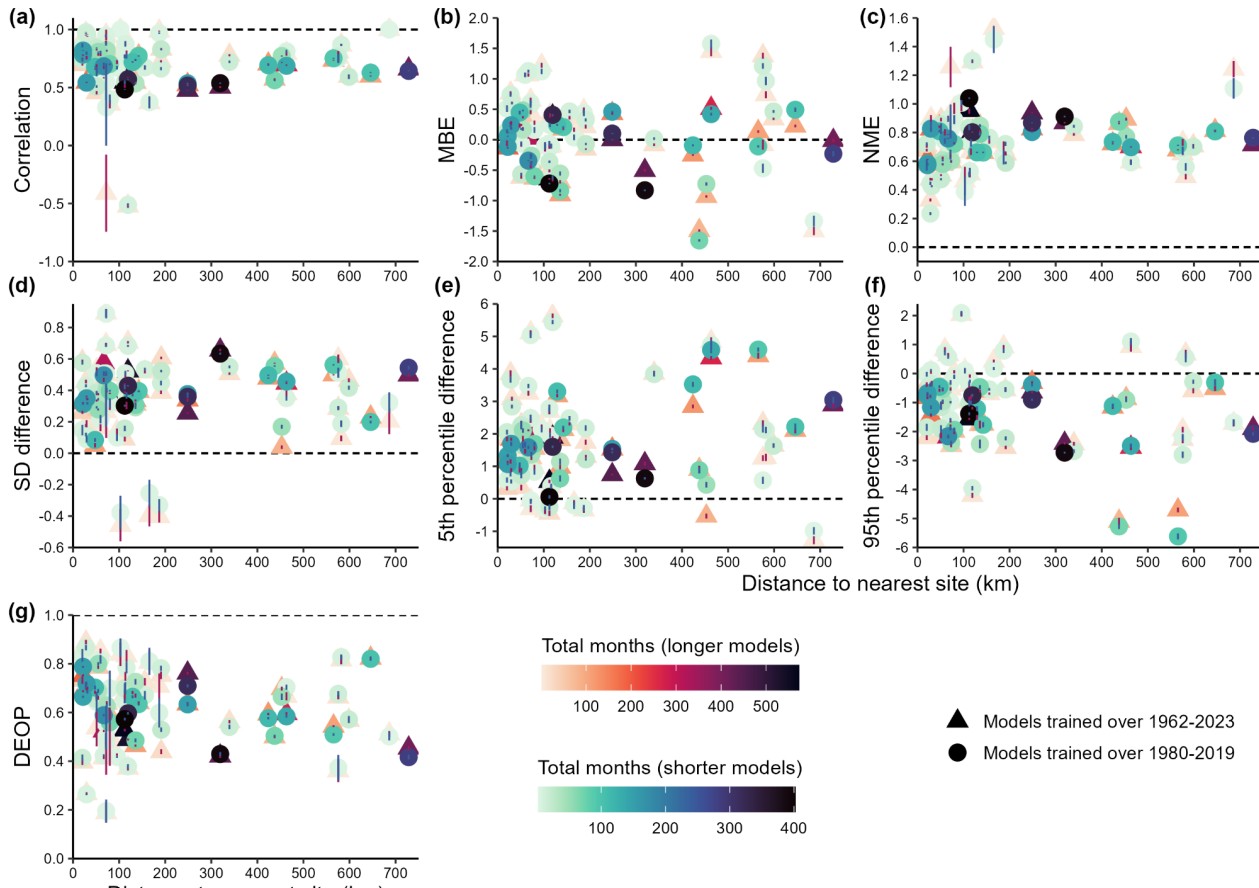

**Figure 1: Crossplots summarising a suite of independent metrics describing out-of-sample random forest model skill in predicting spatial δ¹⁸Oₚ variability.** Scores were calculated by removing one site from the training data (the 'test site'), training a random forest model using the remaining data, then using that model to predict values for the excluded test site. This process was repeated 50 times for each site, each time initialising the random forest model with a different random seed. Each point shows the skill for a single site. Sites are plotted according to the distance to the nearest site included in the training dataset. Each site has two points on each crossplot: triangles show results for δ¹⁸Oₚ predicted using the longer models trained over 1962–2023 using the reduced set of predictor variables (without weather objects). Circles show results for δ¹⁸Oₚ predicted using models trained over 1980–2019 using the full set of predictor variables (including weather objects). Sites are coloured according to record length. The vertical line over each point shows +/- one standard deviation in that skill score across the 50 models. In all cases, the dashed black line shows the expected value if the modelled values perfectly matched observations. Panel (a) shows the Pearson correlation coefficient between the observed and modelled monthly δ¹⁸Oₚ values. Panel (b) shows the Mean Bias Error (MBE). Panel (c) shows the Normalised Mean Error (NME). Panel (d) shows the difference between the Standard Deviation (SD) of the observed and modelled values (following (Abramowitz et al., 2024), a positive value denotes modelled values have lower SD than observations). Panel (e) shows the difference in the 5th percentile of modelled versus observed values (a positive value denotes the 5th percentile of the modelled values has a positive bias relative to observations). Panel (f) shows the difference in the 95th percentile of modelled versus observed values (a negative value denotes the 95th percentile of the modelled values has a negative bias relative to observations). Panel (g) shows the density estimate overlap proportion (DEOP), which summarises the degree of overlap between density estimates calculated for the modelled and observed values (a value of 1 would indicate perfect overlap; 0 would indicate no overlap). See Section 2.2.1 or Table S4 in Abramowitz et al. (2024) for skill metric definitions.



**Figure 2: Maps showing average random forest model skill in predicting out-of-sample δ¹⁸O_P variability at the 60 sites across Australia with δ¹⁸O_P observations.** Scores were calculated by removing one site from the training data (the 'test site'), training a random forest model using the remaining data, then using that model to predict values for the left-out test site. This process was repeated 50 times for each site, each time initialising the random forest model with a different random seed. Points are coloured according to the average skill score from the 50 models, and are sized according to the record length (record lengths range from 2 to 573 months). Values are monthly δ¹⁸O_P predicted using the longer models trained over 1962–2023, using the reduced set of predictor variables (without weather objects). Skill metrics across panels (a–g) are all as per Fig. 1. The average skill score across all sites is shown in the lower left corner.





970

**Figure 3: Violin-and-boxplots ('voxplots') summarising a suite of independent metrics describing model skill in predicting out-of-sample temporal $\delta^{18}O_P$ variability (across all sites).** Scores were calculated by removing a random 10% of all training data, predicting the missing values, then assessing the model's skill in retrieving the held-out data. This process was repeated such that all training data were tested out-of-sample. Additionally, each random forest model was calculated 50 times, each time with a different random seed. These two iterative processes comprise the distributions. In each panel, the left voxplot shows results for monthly $\delta^{18}O_P$ predicted using models trained over 1962–2023, using the reduced set of predictor variables (without weather objects). The right voxplot shows results for monthly $\delta^{18}O_P$ predicted using models trained over 1980–2019, using the full set of predictor variables (including weather objects). Skill metrics across panels (a–g) are all as per Fig. 1.





**Figure 4: Crossplots comparing modelled and observed monthly δ¹⁸Oₚ values at the 60 sites across Australia with δ¹⁸Oₚ observations.** For each site, δ¹⁸Oₚ values were predicted out of sample (i.e., that site's data was removed from the training dataset). Each point shows the mean of values predicted by 50 unique random forest models, each initiated with a different random seed. Sites are arranged by decreasing latitude, with the northern-most site (Darwin) in the top left corner and the southern-most site (Margate) in the bottom right corner. Site details are as per Table S1. In each panel, the thin black line shows the expected relationship of modelled δ¹⁸Oₚ values exactly matched observations (1:1). The blue line shows the linear relationship between the modelled and observed δ¹⁸Oₚ values, with the 95% confidence interval shown in the blue shading. Data points are coloured by month to highlight the seasonal cycle. Values are from the longer models trained over 1962–2023, using the reduced set of predictor variables (without weather objects).







**Figure 5: Average seasonal cycle of δ¹⁸O_P at 60 sites across Australia, in observations (grey); the random forest models described in this paper (blue); and the ECHAM6-wiso simulation nudged to the ERA5 reanalysis (see Methods; green).** Seasonal cycles are calculated from $\delta^{18}O_P$ values in the interval overlapped by all three data sources (1979–2021), and only include months with observations present in all three data sources. In the case of the random forest-predicted $\delta^{18}O_P$, values are from the longer models trained over 1962–2023, using the reduced set of predictor variables (without weather objects), and show the mean of the 50 models created with unique random seeds. Sites are arranged by decreasing number of observations. Brisbane (top left) has the most observations (n = 500); Wilkawatt and Exmouth (bottom right) have the equal fewest observations (n = 3).



**Figure 6: Density plots comparing the distributions of monthly δ¹⁸O$_P$ values at 60 sites across Australia, from observations (grey); the random forest models described in this paper (blue); and the ECHAM6-wiso simulation nudged to the ERA5 reanalysis (see Methods; green).** Density curves are calculated from δ¹⁸O$_P$ values in the interval overlapped by all three data sources (1979–2021), and only including months with observations (i.e., curves are constructed with δ¹⁸O$_P$ values from exactly the same months). In the case of the random forest-predicted δ¹⁸O$_P$, values are from the longer models trained over 1962–2023, using the reduced set of predictor variables (without weather objects), and show the mean of the 50 models created with unique random seeds. Sites are arranged by decreasing number of observations across 1979–2021. Brisbane (top left) has the most observations (n = 500); Wilkawatt and Exmouth (bottom right) have the equal fewest observations (n = 3).





**Figure 7: Average importance rank of each predictor variable with respect to the random forest models' predictive performance, calculated using the 'permutation' method.** The panels show the overall rank of each variable's relative importance in predicting spatio-temporal precipitation isotopic variability across the Australian continent. Ranks are the average for that variable across 50 random forest models, each created with a different random seed. The least important variable for a predictor model receives a rank of 1; the most important variable receives a rank equal to the number of variables included. So, the most important variables for the model are shown at the top of each panel; the least important at the bottom. Predictors are coloured according to the variable type: blue for meteorological variables, lavender for weather objects, forest green for climate indices (representing remote dynamical drivers), black for month of the year, and lime green for elevation. Panel (a) shows results for the $\delta^2H_P$ models trained over 1962–2023, using the reduced set of predictor variables (without weather objects). Panel (b) is as per panel (a) but for $\delta^{18}O_P$. Panel (c) is as per panel (a) but for $dxs_P$. Panels (d–f) are as per panels (a–c) but for the models trained over 1980–2019, using the full set of predictor variables (including weather objects). Variable names and descriptions are as per Table S2.







**Figure 8: Average importance rank of each predictor variable with respect to the random forest models' predictive performance, calculated using the 'permutation' method.** The panels show the overall rank of each variable's relative importance in predicting temporal precipitation isotopic variability at the seven longest precipitation isotopic monitoring stations in Australia. Site details are as per Table S1. Sites are arranged in order of descending latitude from top to bottom. Ranks are as per Fig. 7, with the most important variable receiving the highest rank, and less important variables receiving lower ranks. As per Fig. 7, predictors are coloured according to the variable type; variable names and descriptions are as per Table S2. Here only showing the five most important predictor variables at each site. The first column shows results for $\delta^2H_P$ predicted at the seven sites, with models trained over 1980–2019, using the full set of predictor variables (including weather objects). Second column is as per the first column but for $\delta^{18}O_P$. Third column is as the first column but for $dxs_P$.





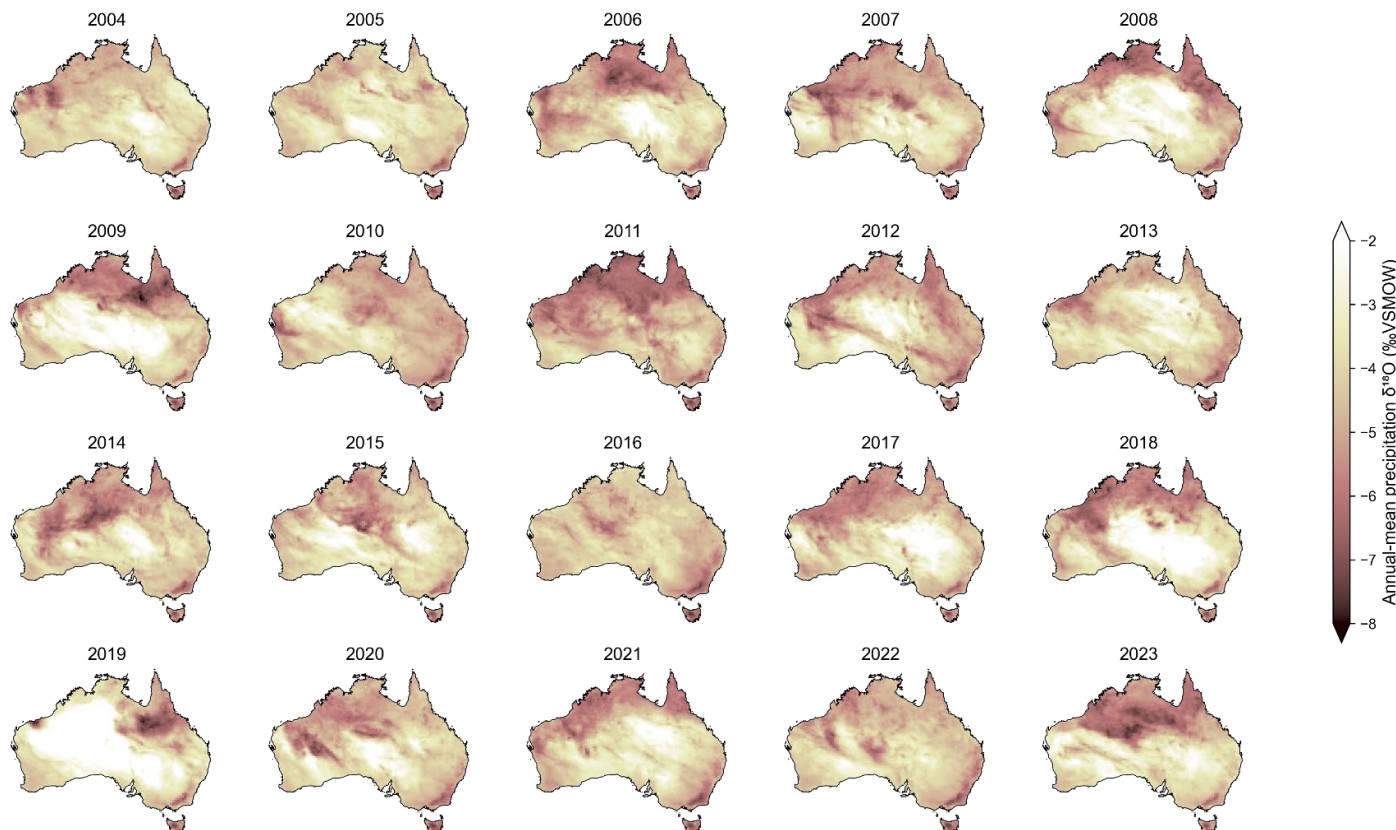

**Figure 9:** Precipitation amount-weighted annual mean $\delta^{18}O_P$ values across Australia for the 20 most recent years of the random forest models trained over 1962–2023, using the reduced set of predictor variables (without weather objects).



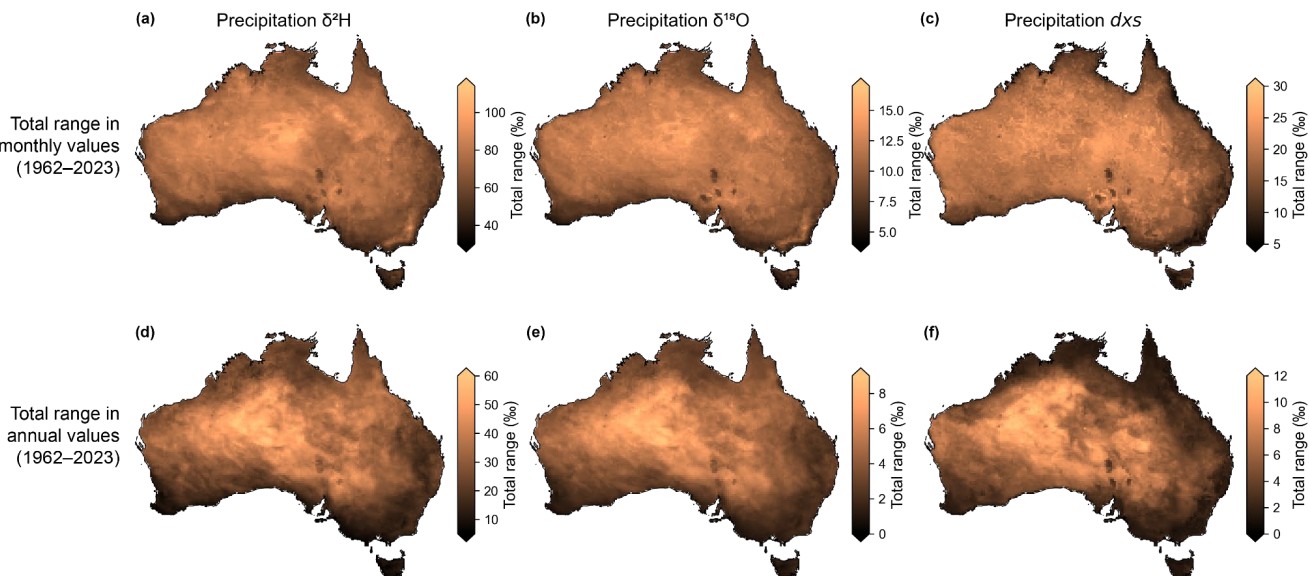

**Figure 10: Maps showing the total range in precipitation isotopic values over the 1962–2023 period.** Maps show the total inter-month (top row) and inter-annual (bottom row) range in $\delta^2H_P$ (first column), $\delta^{18}O_P$ (second column) and $dxs_P$ (third column) values at all locations across the Australian continent. That is, at each location the maps shows the difference between the most positive value occurring in the 1962–2023 period and the most negative value occurring in that period. Ranges calculated using the random forest models trained over 1962–2023, using the reduced set of predictor variables (without weather objects). Annual values are precipitation amount-weighted annual means.





**Figure 11: Long-term annual mean precipitation isotopic variability across the Australian continent ('isoscapes').**
Panel (a) shows long-term (1962–2023) annual-mean precipitation amount-weighted $\delta^2 H_P$ as calculated from the random forest models described in this paper (showing the mean of the 50 models created with unique random seeds). Panel (b) shows long-term (1979–2021) annual-mean precipitation amount-weighted $\delta^2 H_P$ as simulated by ECHAM6-wiso nudged to ERA5 (Cauquoin and Werner, 2021). Panel (c) shows long-term (1962–2014) annual-mean precipitation amount-weighted $\delta^2 H_P$ as estimated by linear regression using geographical variables (Hollins et al., 2018). Panels (d–f) are as per panels (a–c) but for $\delta^{18} O_P$. Panels (g–i) are as per panels (a–c) but for $dxs_P$. Note that the isoscapes from Hollins et al. (2018) were calculated using data spanning 1962–2014, which exceeds the total coverage of the ECHAM6-wiso simulation. As we could not average across matching time periods for all three data sources, we show annual means across each dataset's entire coverage interval to provide the most representative long-term isoscapes. Points show the long-term annual mean precipitation isotopic values at each site with 2 or more years of observations (see Section 2.4 for details). Point sizes scale log-linearly with record length (in years).