# Peer review of "High resolution monthly precipitation isotope estimates across Australia from machine learning"

_EGUsphere, 2025_

## Author Comment (AC2)

Tables and figures accompanying the authors' responses to reviews of

**High resolution monthly precipitation isotope estimates across Australia from machine learning**

Georgina Falster et al.

Correspondence to: Georgina M. Falster (georgina.falster@adelaide.edu.au)

This file contains the following materials:

Table A

Figures A-I

Table A: Relationship of monthly precipitation amount measured at precipitation isotope sampling locations with precipitation amount in the same months as inferred from AGCD v2 and ERA5. Including all sites where precipitation amount was reported alongside precipitation stable isotope composition.

| Site              | Correlation |      | Regression coefficient |      | # months |
|-------------------|-------------|------|------------------------|------|----------|
|                   | AGCD v2     | ERA5 | AGCD v2                | ERA5 | # шоппь  |
| Adelaide          | 0.73        | 0.63 | 1.28                   | 0.54 | 399      |
| Alice Springs     | 0.97        | 0.85 | 1.01                   | 0.87 | 332      |
| Barakula          | 0.83        | 0.71 | 0.81                   | 0.76 | 13       |
| Big Hill          | 0.87        | 0.71 | 0.72                   | 0.42 | 35       |
| Braidwood         | 0.98        | 0.83 | 1.04                   | 0.68 | 16       |
| Brisbane          | 0.95        | 0.82 | 0.85                   | 0.61 | 709      |
| Cairns            | 0.37        | 0.30 | 2.81                   | 1.74 | 1027     |
| Cape Grim         | 0.83        | 0.74 | 0.80                   | 0.66 | 501      |
| Charleville       | 0.96        | 0.85 | 0.87                   | 0.80 | 126      |
| Chillagoe         | 0.60        | 0.57 | 2.07                   | 1.83 | 151      |
| Cobar             | 0.96        | 0.76 | 0.82                   | 0.62 | 119      |
| Darwin            | 0.45        | 0.39 | 0.54                   | 0.47 | 1149     |
| Esperance         | 0.75        | 0.74 | 0.69                   | 0.67 | 20       |
| Evatt             | 0.91        | 0.81 | 0.87                   | 1.12 | 12       |
| Halls Creek       | 0.96        | 0.91 | 1.10                   | 0.96 | 15       |
| Horsham           | 0.92        | 0.56 | 0.87                   | 0.66 | 11       |
| Lithgow           | 0.96        | 0.83 | 1.10                   | 0.88 | 43       |
| Lucas Heights     | 0.24        | 0.18 | 0.65                   | 0.40 | 889      |
| Macquarie Marshes | 0.35        | 0.26 | 0.97                   | 0.72 | 284      |
| Margate           | 0.97        | 0.90 | 0.83                   | 0.57 | 18       |
| Meekatharra       | 0.98        | 0.73 | 0.90                   | 0.74 | 119      |

Figure Aa: Maps showing the consequences of modelled  $dxs_P$  across the Australian continent directly versus calculating  $dxs_P$  from modelled  $\delta^2H$  and  $\delta^{18}O$ . First panel shows long-term annual mean precipitation dxs across the Australian continent, modelled directly as described in Section 2.2 (and shown in Fig. 11g). Second panel shows long-term annual mean precipitation dxs across calculated from the random forest-modelled  $\delta^2H_P$  and  $\delta^{18}O_P$  as follows:  $dxs_P = \delta^2H_P - 8 * \delta^{18}O_P$ . Third panel shows the difference between  $dxs_P$  as estimated by the two approaches; the  $dxs_P$  values implied by the separate  $\delta^2H_P$  and  $\delta^{18}O_P$  models subtracted from the directly-modelled  $dxs_P$  values.

Figure Ab: As per Fig. Aa but showing results for each individual month. Top row shows directly-modelled  $dxs_P$ , middle row shows  $dxs_P$  as calculated from the random forest-modelled  $\delta^2H_P$  and  $\delta^{18}O_P$ , and bottom row shows the difference between the two, calculated as described in Fig. Aa.

Figure B: Standard deviation of random forest-modelled monthly (rows one to four) and annual (row five) precipitation isotopic variability, across the 50 random forest model ensemble members. Standard deviations calculated on the long-term monthly mean (rows one to four) or precipitation-weighted annual mean (row five), across the 50 models created with unique random seeds. Here showing only the central month of each season. Ensemble standard deviations calculated using the random forest models trained over 1962–2023, using the reduced set of predictor variables (without weather objects).

Figure C: Comparison of monthly precipitation amount (1962–2023) estimates from the Australian Gridded Climate Dataset v2 (AGCDv2; Evans et al., 2020) and the European Centre for Medium-Range Weather Forecasts Reanalysis v5 (ERA5; Hersbach et al., 2020). Comparison is shown for each precipitation isotope monitoring sites used in this study; the data were extracted from the grid cell in each product closest to the site. In each panel, the thin black line shows the expected relationship if precipitation in AGCDv2 and ERA5 match exactly (1:1). The blue line shows the linear relationship between AGCDv2 and ERA5 precipitation.

Figure D: Plots showing the difference between modelled and observed monthly  $\delta^2 H_P$  values at the 59 sites across Australia with  $\delta^2 H_P$  observations. For monthly observation at each site, the observed  $\delta^2 H_P$  value was subtracted from the equivalent random forest-modelled  $\delta^2 H_P$  value. Sites are arranged by decreasing latitude, with the northern-most site (Darwin) in the top left corner and the southern-most site (Hobart) in the bottom right corner. Site details are as per Table S1. Data points are coloured by month to highlight the seasonal cycle.

Figure E: As per Figure D but for  $\delta^{18}O_P$ .

Figure F: As per Figure D but for dxsp.

Figure G: Violin-and-boxplots ('voxplots') summarising bias in modelled monthly  $\delta^2 H_P$ ,  $\delta^{18}O_P$ , and  $dxs_P$  values with respect to observed values at the equivalent months/locations. Each distribution summarises all months at all locations for each decade. Modelled values are from the longer models trained over 1962–2023, using the reduced set of predictor variables (without weather objects), and in all cases are the mean of values predicted by 50 unique random forest models, each initiated with a different random seed.

Figure H: As per Fig. G but with bias values summarised by both latitude and season.

Figure I: Difference in long-term precipitation amount-weighted annual mean precipitation isotopic variability across the Australian continent as estimated by three methods. The first column shows the annual precipitation isotope climatology for  $\delta^2 H$ ,  $\delta^{18} O$ , and dxs, as calculated from the random forest models described in this paper. The second column shows the difference between the annual precipitation isotope climatology in the random forest estimates and those of ECHAM6-wiso nudged to ERA5 (Cauquoin and Werner, 2021). The third column shows the difference between the annual precipitation isotope climatology in the random forest estimates and those estimated by linear regression using geographical variables (Hollins et al., 2018). Both the ECHAM6-wiso and linear regression isoscapes were regridded to match the spatial resolution of the random forest models using a mass-conservative regridding scheme. After regridding, the ECHAM6-wiso and linear regression estimates were subtracted from the random forest estimate to produce the difference maps shown. Note that the three isoscape climatologies for each metric were calculated using data spanning different time periods (see Fig. 11 caption for details). Points in the first column show the long-term annual mean precipitation isotopic values at each site with 2 or more years of observations (see Section 2.4 for details). Point sizes scale log-linearly with record length (in years).

---

## Author Response (AR1)

Dear Dr Orlowski,

Many thanks for your positive assessment of our manuscript, and we were very pleased to read such positive and enthusiastic comments from both Reviewers.

We have implemented all Reviewer suggestions as per our online replies to the Reviewers. Carefully considering your suggestion, we added only a single new supplementary figure, which is not central to the manuscript's findings, and is likely to be of interest to relatively few readers. Otherwise, we stuck to what was really needed to address the Reviewers' requests for extra details.

In summary, we have
- Added two new main-text figures
  - Figure 4, summarising model bias by decade
    - addressing a query from Reviewer 2 on whether model skill changes through time
  - Figure 8, comparing our directly-modelled *dxs* values with *dxs* calculated from the modelled $\delta^2$H and $\delta^{18}$O values
    - addressing a suggestion from Reviewer 1 to show how the *dxs* values implied by the separate $\delta^2$H and $\delta^{18}$O models deviate from the predictions of the *dxs* model
- Added extra panels to one existing main-text figure
  - Figure 13, which now directly compares the random forest $\delta^2$H, $\delta^{18}$O, and *dxs* climatologies with those from alternate sources (ECHAM6-wiso, linear regression models)
    - addressing a request from Reviewer 2 to provide these difference maps
- Added one new supplementary figure
  - Figure S7, showing the standard deviation across the 50-member ensemble of predictions from the random forest models calculated with unique random seeds
    - addressing a request from Reviewer 2 to include these maps in the supplement

In addition, the Zenodo repository containing all output from this paper is now live (accessible from https://doi.org/10.5281/zenodo.15486277) and indeed data described in this paper has already been downloaded 438 times. The web app where users can extract timeseries data is also live (available from https://wateriso-aus.shinyapps.io/apic).

Please see below for point-by-point responses to the Reviewers' comments. Original reviewer comments are in red, our responses directly to the reviewers are in *italicised black* (these are exactly as per our publicly available online replies), and implemented changes are described in **bold black**. All line numbers correspond to the version of the manuscript with tracked changes.

Regards,

Georgy Falster

- Falster et al. model a large precipitation isotope dataset from Australia using Random Forest, compare the results to those from two other methods, and present and interpret a set of historical monthly average precipitation isoscapes. This is an excellent study, and very well presented…in many ways it's the precipitation ML isoscape study I've been hoping to see for several years now! It's thorough, really uses the power of the method and an expanded suite of features to go well beyond what's been done with other statistical methods and learn more from our isotope data. In so doing it represents one of the first successful attempts at a data-driven, time-explicit analysis of historical precipitation isotope patterns. Kudos to the authors, and I'm excited to see this published.

*Many thanks for this!! We spent a lot of time trying to do this as well as possible with the tools to hand, so it was really great to read this assessment from someone who has done a lot of work on this topic over many years.*

- Below are a handful of comments and suggestions that I hope might be useful and help the authors tie up a few loose ends.

*Thanks—these are all super-helpful suggestions and we have addressed each below.*

- I have mixed feelings about the choice to model D-excess directly. I've done this, too, and am not fundamentally opposed to this approach. But it does lead to a fundamental inconsistency…because D-excess is not and independent parameter you have 3 independent models that are describing a system with only two degrees of freedom. In an ideal world, this would be modeled as a multivariate system, since H and O isotopes have a lot of shared information. A single self-consistent model could be fit to simultaneously predict δ2H, δ18O, and from them D-excess. Maybe a next step, but in the current manuscript it would at least be interesting to see how strongly the D-excess values implied by the separate δ2H and δ18O models deviate from the predictions of the D-excess model. Areas w/ large differences imply inconsistency in the models, which could be due to the influence of specific (poorly represented) forcing factors, incomplete or inconsistent data, or other factors that might motivate future work.

*Agreed—I did waver a bit on this point before, in the end, separately modelling the precipitation deuterium excess. In fact, I did already perform this comparison, but didn't include it in the original submission. This was in part for fear of overwhelming the readers with too enormous a supplement, but also because I suspect there is quite a lot of 'science' in that comparison that could potentially merit a longer discussion. But ultimately yes—it will be good for users of these isoscapes to have that assessment to hand. And that's a fun suggestion to simultaneously model $\delta^2H$ and $\delta^{18}O$—something for future work. It is indeed clear across all the results that when using the random forest approach, as expected from the isotope physics, H and O have a very high amount of shared information across predictors.*

*In any case, the comparison is in the attached file (Fig. A), and will be included in the revised manuscript. Pleasingly, Fig. Aa shows that there is minimal difference in long-term annual mean precipitation dxs when modelled directly versus when calculated from modelled $\delta^2H_P$ and $\delta^{18}O_P$. Broadly,*

*the independent random forest model predicts slightly higher $dxs_P$ across inland western Australia than implied by the $\delta^2H_P$ and $\delta^{18}O_P$ models. The reverse is true for the southern and northern coastline, excepting the far-northern tips of the continent. However, the magnitudes of the difference are small in the long-term annual mean.*

*The differences are slightly larger for the monthly climatologies, where there is a seasonal cycle in the difference between directly-modelled $dxs_P$ versus $dxs_P$ calculated from $\delta^2H_P$ and $\delta^{18}O_P$ (Fig. Ab). The difference is biggest in January, where the independent random forest model predicts higher $dxs_P$ across most of the continent than the $\delta^2H_P$ and $\delta^{18}O_P$ models. The difference is less for most other months, except around the south-western coastline in December where the reverse is true.*

*Our sense is that digging more deeply into this would be beyond the scope of the current manuscript, however we are very grateful for the opportunity to provide this extra information for the readers, and look forward to investigating the reason for the discrepancy in more detail. We would most likely include Fig. Aa in a revised version of the manuscript, with the extra detail provided by Fig. Ab available here for interested readers.*

**We have added a comparison of directly-modeled $dxs_P$ with $dxs_P$ as calculated from $\delta^2H_P$ and $\delta^{18}O_P$. Text describing this comparison has been added to the Methods at L189-191 and Results at L401-406. We have also added a new figure to the main text that shows the results of this comparison (Fig. 8).**

- Methods: Did you make any attempt at feature selection? I realize this is less important for RF than for many other methods but can still be beneficial. The very smooth decline in feature importance in Fig. 7 is interesting to me and could reflect some influence of highly correlated features. I think it would be work checking/reporting on this, at least.

*We did not perform feature selection, instead relying on 1) a careful initial choice of predictors; then 2) the random forest's predictor importance algorithm to determine if any predictors were detrimental to model skill (not the case here). In fact, we chose the random forest method because precipitation isotopic variability is influenced by so many highly correlated variables, and we were hoping to capture as much of the nuance across those relationships as possible. We will add a statement to this effect to the Methods section (2.3 Predictor Variables) of the revised manuscript.*

**We have added this statement to the Methods at L296-301.**

- In several places you refer to D-excess as an 'isotope system' (e.g., line 349, 398, others)…which isn't quite correct, it's a derived parameter that integrates information from two isotope systems. I suggest adjusting your terminology for correctness. For example you could refer to 'three isotopic metrics' instead of 'three isotope systems'.

*Yes we dithered for quite a while on this terminology! And couldn't find anything consistent across the literature. But we will change the wording to 'isotope metrics' in all instances.*

**We have changed this terminology in all cases ('isotope systems' → 'isotope metrics').**

- L 350-354: this is an important point given RF's inherent inability to meaningfully extrapolate beyond the training data's feature space. Thank you for including this information.

*No worries (: we agree that this is a very important point, and something that is probably not widely known about the Australian continent's climatic feature space—that actually it can be captured reasonably well from a small subset of locations.*

- L 428, also previously: The text implies that the term 'isoscape' refers specifically to climatology (long-term average models), which is not the case – the term has been applied to space- and/or time-explicit models of isotopic variation since its inception (e.g., Bowen, West, & Hoogewerff, 2009; Bowen, West, Vaughn, et al., 2009; West et al., 2010).

*Thanks for the clarification and references—we will correct the terminology to simply differentiate between space-and-time-explicit isoscapes and their climatologies.*

**Throughout the manuscript, we have adjusted our terminology to differentiate between space-and-time-explicit isoscapes and their climatologies ('time-mean isoscapes').**

- L 580-581: This line in the data availability statement is unclear – are the data themselves also available in the Zenodo archive referenced in the previous section? If so, please clarify, if not, please indicate where they are available.

*The new random forest isoscapes are all now publicly available from the Zenodo archive (i.e., the link is now live - here it is again for reference https://zenodo.org/records/15486278). Things are more complicated for the underlying observational data used to train the models. The availability of those data is outlined in Table S1, and we will add that reference to the Data Availability statement to avoid confusion. But In brief: I used a combination of published data (e.g., GNIP), unpublished datasets obtained directly from the authors (who are co-authors in all cases), and one dataset that is published but not freely available (Yarrangobilly, Tadros et al. 2022 QSR).*

*In summary: some of the underlying observational data are available from online repositories (outlined in Table S1), but some are as yet unpublished or not available in online archives.*

**We have added a statement to the Data Availability section pointing to Table S1 which describes the availability of the underlying observational data.**

- Fig 5: Symbology could be adjusted to make it a little easier to distinguish the different series…the differences are quite subtle and hard to pick out on the small panels.

*Thanks for this suggestion—we will update Fig. 5 and its equivalent supplementary figures to differentiate the timeseries with symbols as well as the line colours.*

**I have updated Fig. 5 (now Fig. 6) and the equivalent supplementary figures (Figs. S11-12) so the three series are differentiated both by colour and symbol.**

**RC2**

- This is a well-executed paper demonstrating a rigorous application of machine learning method to expand the Australian stable water isotope record. The results are clear and consistent. The combination of model testing, predictor diagnostics and spatial coverage makes it a valuable dataset to the community. My comments below are intended to strengthen the interpretation and presentation, however, the paper is already very strong overall. Below are my comments:

*Many thanks for the positive review, and we too hope for this to be a valuable community asset. Thanks also for the great suggestions—we have addressed each below, and provide all new analyses in the attached document.*

- The authors say that each isotope model was trained 50 times with random seeds. Figures 1-3 already present site-level uncertainties across the 50 random-forest runs. These quantities show how stable these model skill parameters are relative to observations. However, understanding the uncertainties within the predicted isotope fields themselves would be useful too. For this, the authors could include a map of ensemble spread in the supplement. For example, the standard deviation of dD, d18O and dxs predictions across all the 50 runs at each grid cell would be useful to highlight where model confidence is low or high.

*We had in fact performed this analysis, but did not include it in the original submission for fear of overwhelming readers with too many supplementary figures(!) However, we agree that it is useful information for isoscape users, and will include the maps of ensemble spread in a revised manuscript. Specifically, for each precipitation isotope metric ($\delta^2H_P$, $\delta^{18}O_P$, $dxs_P$) we show the standard deviation across the 50-member ensemble, for the long-term annual mean as well as the centre months of each season (see Fig. B in the attached file, which will be included in the supplement of a revised manuscript). As shown on Fig. B, the magnitude of variability across the 50-member ensemble is very small—highlighting the stability of the models.*

**We have added this figure to the Supplement (Fig. S7) and referenced it in the Results (L365).**

- The manuscript relies on ERA5 for meteorological predictors, including precipitation amount and intensity. While the authors mention that ERA5 and AGCDv2 precipitation show "very similar" results, this statement could benefit from quantitative support. Because precipitation amount and intensity are among the dominant predictors in the isotope models, any biases in ERA5 are likely to propagate into the isotope estimates. I recommend that the authors quantify ERA5 against AGCDv2 using a scatterplot or bias map of monthly precipitation at isotope sites (or even a table would do).

*This is another analysis that we had already performed but weren't sure whether or not to include given the already-long supplement. But again, we agree that readers may be interested in this point, especially*

*given there is a bit of nuance to it. We therefore provide more information here so it is accessible to readers (as well as an additional figure and table), and will also add this discussion to the revised manuscript if requested by the Editor.*

*In the attached file, Fig. C compares monthly precipitation from the AGCDv2 and ERA5, across every month from 1962–2023 (the time span of the long isoscapes) at each monitoring site. The thin black line shows a 1:1 relationship. At some sites, precipitation estimates from the two products are extremely similar. At others, the slope is flatter than 1, implying that ERA5 produces slightly too little precipitation at the high end.*

*However, there is some nuance to this. First (and most importantly), the sites with the ERA5~AGCDv2 slope closest to 1 are not necessarily the same sites where the modelled precipitation $\delta^2H/\delta^{18}O/dxs$ is the best match for observed precipitation $\delta^2H/\delta^{18}O/dxs$—including with respect to extremes (e.g., Fig. 4 in the original manuscript [now Fig. 5]). Second, precipitation estimates from neither ERA5 nor AGCDv2 are a perfect match for the precipitation amount observations recorded alongside the precipitation isotopes. Table A shows all precipitation isotope monitoring sites that also reported precipitation amount. For each, we show a) the correlation, and b) the regression coefficient (slope) with respect to precipitation from AGCDv2 and ERA5. The average correlation of observed precipitation with AGCDv2 precipitation is higher than that with ERA5 precipitation, however the regression coefficients for AGCDv2 ~Obs tend to be >1, whilst the regression coefficients for ERA5~Obs tend to be <1. This discrepancy increases when using only sites with >100 observations.*

*This is a point that, in fact, tends to be glossed over in many studies relying on interpolated data or reanalysis products—there are uncertainties even in observational products. We acknowledge that this is an uncertainty we did not explicitly account for (e.g., by obtaining all predictor variables from multiple sources and doing all other method steps several times accordingly), but we considered that with all the other uncertainties incorporated into the method, this would make the modelled and uncertainty quantification processes quite unwieldy. We also considered that this would not make a major difference to the results, although we acknowledge that this could be a contributing factor to the under-estimated extreme precipitation $\delta^2H/\delta^{18}O/dxs$ values (as already stated at L474-475 in the original manuscript). In any case, we hope that the new plots here will be of interest to some readers (for reasons beyond just this study, too!) and thank the Reviewer for the opportunity to include them (:*

**As per this response, we stated already in the text that we compared AGCD precipitation with ERA5 precipitation (L262-264) and state that the daily values required to calculate precipitation intensity are not available from the AGCDv2 (L261). Given the above detailed discussion required to contextualise the additional figures is quite long and requires the support of extra supplementary figures and tables, we propose leaving this longer discussion here in the publicly-available review rather than adding it into the text. As stated in the reply to the Reviewer, we already provide the relevant summary statement of the implications of using ER5 precipitation data at L512-514.**

- The lowest model skill occurs for dxs, due to its linkage to moisture-source humidity and temperature. The authors acknowledge that these source conditions are not directly represented and that when "weather objects" are introduced, there is some partial improvement in the model.

While a full trajectory modeling would indeed by very computationally expensive, the paper could still test or discuss some simpler methods that could capture the source-region variabilities (e.g., ERA5 based upwind SST or column-integrated humidity gradients, etc.). Even a short comparison between the dxs skill improvements with and without the introduction of weather objects would be useful to clarify how users should be aware of the missing source information for the data usage.

*Regarding 'Even a short comparison…': At L345-347 we state that $dxs_P$ is the only isotope metric for which the addition of the weather objects results in a major increase in a particular skill metric (the density estimate overlap proportion). We also state at L486-490 "The shorter models—incorporating the weather object data—predicted $dxs_P$ more skillfully than the longer models without the weather objects. Further, the skill increase resulting from inclusion of the weather objects was larger for $dxs_P$ than for $\delta^2H_P$ and $\delta^{18}O_P$, suggesting that the moisture source and transport information inherent in the weather objects is particularly important for $dxs_P$".*

*Regarding the suggestion that we could discuss some simpler methods for capturing the moisture-source conditions: the Reviewer accurately summarised that we tried to do this as comprehensively as possible with the inclusion of the weather objects—which in itself is a major step forward in isoscape calculation as this data type has not been used in any previous isoscape studies. However, considering this same point we did also test the effect of including the vertically-integrated water vapour flux as a predictor. The addition of this variable did not result in any skill increase (or indeed any change in the results), which suggests that the information was likely captured intrinsically in the other meteorological variables.*

*The reviewer's suggestion to use ERA5-based upwind SST would run into the same problems as the back-trajectory modelling (outlined in the manuscript at L491-494). That is, it would be extremely computationally expensive to identify the relevant regions for the upwind SST for all grid cells for all months. However, it would be a very interesting avenue of future research to use this information to model $dxs_P$ at a single location (or small geographical region), and we plan to do this in the coming years.*

**As per the above response provided to the Reviewer, we do not consider that there are necessary changes to the manuscript arising from this comment.**

- In section 2.2.1, the authors explain that they test temporal transitivity by randomly leaving out 10% of all observations and using the rest to train the model. This random sampling is effective in checking how well the model predicts data that look similar to what it has already seen. However, it does not give much information on how the model will perform for changes over time. Since Australia's climate has shifted over the last several decades, it would be helpful to see how stable the model's performance has been over time. For this, I suggest that the authors add a simple figure showing the model error or correlation changes per decade, or simply plot residuals in a time series. These tests will help users understand whether the model's relationship between isotope and climate can stay consistent over the entire record.

*Thanks for this suggestion—we have performed both of these additional skill tests. The results are shown in the attached document, and we will add these to the supplement of a revised manuscript along with a brief discussion in the Results (Section 3.1).*

*Fig. D shows, for all sites, the difference (total difference rather than residuals from a model) between observed and modelled $\delta^2H_P$ for all months that have observations. The black line shows a perfect match between the observed and modelled values. A positive offset means the modelled $\delta^2H_P$ value is too high relative to the observed value and vice versa for a negative offset. Figs. E and F show the same for $\delta^{18}O_P$ and dxs$_P$, respectively. The plots suggest that model performance is fairly stable though time.*

*The same is evident from Fig. G, which summarises the bias in each isotope metric by decade (voxplot widths are scaled by the number of observations in that decade). Again, there is no major change through time in the model bias relative to observations (accounting for data density).*

*Finally, for anyone looking for more information on temporal variability in model skill, we have also included Fig. H, which shows model bias by season, plotted in 5° latitude bins.*

**We have added Fig. G from our response file as a new main-text figure (Fig. 4), along with text describing the figure in the Methods (L218-221) and Results (L372-373). We consider that of the two options suggested by the Reviewer, this is the figure that best (and most concisely) addresses the Reviewer's request for more details on the temporal stability of the model.**

- The RF models are trained on 60 sites located in coastal and near populated regions. The predictive skills show no relationship with distance to the nearest site and that more than 99% of predictor values fall within the training range. While this confirms good coverage in predictor space, the manuscript will gain from a quantification over the unobserved inland regions. The authors could provide a difference map between RF and ECHAM6-wiso climatology across inland Australia.

*We agree that this is a tricky point, and a lack of inland monitoring data is not just an issue for precipitation isotopes in Australia, but for many climate variables—even including precipitation amount. The lack of observational data across much of the continent was a main motivator for this study, although I acknowledge that it does make both verification and accurate uncertainty quantification very difficult. Unfortunately there is currently no more data available for further model verification inland than we have already used in this study.*

*In any case, we have created the difference maps as suggested (Fig. I). Broadly, the random forest isoscapes predict higher inland climatological $\delta^2H_P/\delta^{18}O_P$ values than ECHAM6-wiso; the reverse is true for dxs$_P$. It is difficult to say which is more accurate—Fig. S17 shows that the random forests tend to overestimate inland $\delta^2H_P/\delta^{18}O_P$ values with respect to observations, but ECHAM6-wiso tends to underestimate them. These differences may be influenced by bias in the observational training data. Inland Australia is climatologically hot and dry, and is sparsely populated. Because of this sparse population, in many cases data are collected via composite samplers rather than daily rainfall collection, despite the dry climate. Composite samplers tend to be associated with isotopic bias in low-rainfall*

*months, resulting in a positive (but not systematic) bias in $\delta^2H_P/\delta^{18}O_P$ values. Accordingly, ECHAM6-wiso—which directly simulates processes important for inland precipitation $\delta^2H/\delta^{18}O/dxs$, such as subcloud evaporation (Crawford et al., 2017)—underestimates inland precipitation $\delta^2H/\delta^{18}O$ with respect to observations, but may in fact be closer to the true values. We will add a statement to this effect to a revised manuscript.*

*Nevertheless, as stated in Table 1 (and apparent visually from Fig. S17 for the inland sites), the overall magnitude of the (apparent) bias in the random forest isoscapes is less than that of ECHAM6-wiso, lending confidence to our results. This is essentially impossible to test further without new observational data—which we strongly advocate for whilst recognising the expense and difficulty of long-running observational campaigns.*

*Crawford, J., Hollins, S., Meredith, K., Hughes, C.: Precipitation stable isotope variability and subcloud evaporation processes in a semi-arid region, Hydrol. Proc., 31, 20–34, 2017.*

**We have combined Fig. I from our response file with the original main text Fig. 11 (now Fig. 13). We have also added the above discussion to the manuscript text (L555-567).**

- Section 4.3 presents useful examples showing how predictor importance varies by region, but the discussion could be expanded by mentioning why specific predictors dominate in each climate regime and what they imply isotopically. For example, how the high influence of precipitation amount and intensity in the Australian tropics reflects the stronger rainout or amount effect behavior and how to interpret it isotopically. Extending each regional examples in this way would help show how RF predictors reproduce physically meaningful isotope-climate linkage that were introduced in the Introduction.

*We deliberately kept this section short, recognising that this paper largely focuses on describing the methodology and results rather than a synthesis of the climatic drivers of precipitation stable isotopic variability across the Australian continent (which itself is enough for several papers!). Similarly, out of all the sites with enough observational data to carry out the predictor importance assessment, only Darwin is far enough north that we might expect a classic 'amount effect'. However, we acknowledge that conceptual links to the introduction might be missing for some readers. We will add those links in a revised version of the manuscript—for example, touching briefly on the different broad mechanisms by which the various predictor variables might be linked to temporal precipitation $\delta^2H/\delta^{18}O/dxs$ variability in different places (e.g., tropical amount effect, seasonal moisture source changes, local RH etc).*

*For those interested in getting properly into the weeds on this point: we have a follow-up paper well underway which dives into the dynamical drivers of precipitation isotopic variability across the continent in far more detail. I think that it would not be possible to do that justice in a single discussion section here, although I agree that a clearer tie to L59-62 of the Introduction would improve Section 4.3 and will revise this section accordingly.*

**We believe that the above comment sufficiently addresses the Reviewer's comment.**